# Insight into dynamic and steady-state active sites for nitrogen activation to ammonia by cobalt-based catalyst

Xiuyun Wang [1], Xuanbei Peng[1], Wei Chen[2], Guangyong Liu[1], Anmin Zheng [2]*, Lirong Zheng[3], Jun Ni[1], Chak-tong Au[1] & Lilong Jiang[1]*

The industrial synthesis of ammonia ($NH_3$) using iron-based Haber-Bosch catalyst requires harsh reaction conditions. Developing advanced catalysts that perform well at mild conditions (<400 °C, <2 MPa) for industrial application is a long-term goal. Here we report a Co-N-C catalyst with high $NH_3$ synthesis rate that simultaneously exhibits dynamic and steady-state active sites. Our studies demonstrate that the atomically dispersed cobalt weakly coordinated with pyridine N reacts with surface $H_2$ to produce $NH_3$ via a chemical looping pathway. Pyrrolic N serves as an anchor to stabilize the single cobalt atom in the form of $Co_1$-$N_{3.5}$ that facilitates $N_2$ adsorption and step-by-step hydrogenation of $N_2$ to *HNNH, *NH-$NH_3$ and *$NH_2$-$NH_4$. Finally, $NH_3$ is facilely generated via the breaking of the *$NH_2$-$NH_4$ bond. With the co-existence of dynamic and steady-state single atom active sites, the Co-N-C catalyst circumvents the bottleneck of $N_2$ dissociation, making the synthesis of $NH_3$ at mild conditions possible.

[1] National Engineering Research Center of Chemical Fertilizer Catalyst, Fuzhou University, 350002 Fuzhou, Fujian, China. [2] National Center for Magnetic Resonance in Wuhan, State Key Laboratory of Magnetic Resonance and Atomic and Molecular Physics, Key Laboratory of Magnetic Resonance in Biological Systems, Wuhan Institute of Physics and Mathematics, Innovation Academy for Precision Measurement Science and Technology, Chinese Academy of Sciences, 430071 Wuhan, China. [3] Institute of High Energy Physics, Chinese Academy of Sciences, Beijing, China. *email: zhenganm@wipm.ac.cn; jllfzu@sina.cn

NH$_3$ is vital for the production of artificial fertilizer[1,2]. Over Fe-based-derived catalysts, the operation conditions of the Haber–Bosch process are harsh (400–600 °C, 20–40 MPa), and the energy input inevitably large[3]. The dissociation of the extremely stable N≡N triple bond (945 kJ/mol) is the bottleneck[4]. Significant efforts have been put in to develop high-performance catalysts to lower the dissociation energy of N$_2$, but only a few can bypass the bottleneck via shifting the sluggish N$_2$ dissociation to the formation of NH$_x$ (where $x = 0$–3)[1,4,5]. Through theoretical studies, Liu et al.[6] demonstrated that if metal Fe is in the form of individual Fe$_3$ clusters rather than in groups of C7 sites, there is N$_2$ hydrogenation rather than direct N$_2$ dissociation that follows the Langmuir–Hinshelwood (L–H) mechanism. The indirect dissociation of the weakened *N-NH$_2$ bond occurs much more easily following the Eley–Rideal (E–R) mechanism. Nevertheless, there is no experimental evidence to validate that there is preferential hydrogenation of *N$_2$ to *N-NH$_x$ ($x = 2$ or 4, and "*" means adsorbed) except those of the cobalt molybdenum system[7]. It is not easy to investigate the E–R mechanism over traditional catalysts because the reported active sites of Ru- and/or Co-based catalysts for NH$_3$ synthesis exist in the form of nanoclusters or nanoparticles, and the cooperation of five metal atoms[8] makes the energy barriers of L–H mechanism smaller than those of E–R mechanism[6].

The development of single-atom catalysis (SAC)[9–12], in which catalytically active metal is exclusively dispersed as single atoms, provides a good entry to make N$_2$ dissociation impossible, while N$_2$ hydrogenation feasible. It was recently reported that the coordination of transition metals with N could result in dramatic modification of surface electronic properties, such as negative charge density and $d$-band center as well as electron transfer to the antibonding $p$-orbitals of N$_2$[13]. According to Catlow and coworkers[14], the presence of surface N defects on cobalt molybdenum catalysts could enhance NH$_3$ synthesis activity at mild conditions. In general, the binding of a catalyst to N species should not be too strong to avoid catalyst poisoning or too weak to deny activation by H$_2$[15]. Taking into consideration the positive factors of single atoms and surface N defects as well as the excellent adsorption ability of graphitized carbon materials, we synthesized carbon supported N-anchored Co single atom catalyst of high BET surface area to bypass the bottleneck of N$_2$ dissociation, and achieved superior NH$_3$ synthesis activity under mild conditions. On the other hand, it is pivotal to clarify the structure of the single-atom active sites and to identify the related N-involved pathway for the design of catalysts that are efficient for NH$_3$ production at mild conditions. Currently, direct identification and quantitative description of such N-involved active sites under realistic conditions are lacking. With this in mind, we proceeded to investigate the dynamic of single atom sites during NH$_3$ synthesis by a suite of in situ experimental techniques.

Here we show a nitrogen-anchored Co single-atom catalyst (Co–N–C) with dual active sites for efficient and stable NH$_3$ synthesis under mild conditions for the first time, to the best of our knowledge. The single atom Co-based catalyst shows high NH$_3$ synthesis rate of 116.35 mmol$_{NH3}$ g$_{Co}^{-1}$ h$^{-1}$. The use of high-angle annular dark field (HAADF) imaging of aberration-corrected scanning transmission electron microscopy (AC-STEM), operando X-ray absorption spectroscopy (XAS), and in situ X-ray photoemission spectroscopy (XPS), together with $^{15}$N$_2$ isotopic-labeling experiments and theoretical simulations, allows us to answer two fundamental questions: (1) What is the intrinsic nature of the active sites on single atom Co–N–C under NH$_3$ synthesis conditions? (2) Does N in the case of Co–N–C take part in the NH$_3$ production? If so, what kind of N species can participate in NH$_3$ production and how to recycle the N species?

Our studies reveal that the atomically dispersed Co that is weakly coordinated with pyridine N reacts with adsorbed H$_2$ to form NH$_3$, leaving behind an anionic N vacancy. Then, the catalyst can be replenished with N species through the adsorption of gas-phase N$_2$. In other words, the generation of NH$_3$ is via a chemical looping pathway, revealing a reliable process in which single Co sites in the form of dynamic Co–N$_x$ ($0 < x \leq 1.5$) are involved in NH$_3$ production. On the other hand, the single Co sites in the form of steady-state Co$_1$–N$_{3.5}$ are active for N$_2$ adsorption and hydrogenation, as well as for the subsequent formation of NH$_3$ via the breaking of *NH$_2$–NH$_4$ bond following the E–R mechanism.

## Results

**Confirmation of cobalt atomic dispersion.** According to the results of inductively coupled plasma atomic emission spectroscopy (ICP-AES) and element analyses (Supplementary Table 1), the loading of Co is 3.73 wt% and the N/Co molar ratio is 6.04, and the Co−N coordination number (CN) is 5 as confirmed by EXAFS; and the sample is herein designated as Co–N–C. The surface N/Co atomic ratio of Co–N–C based on XPS data is 6.10 (Supplementary Table 1), which is close to the bulk value, implying homogeneous distribution of Co throughout the catalyst. To explore the specific role of single Co atoms, Co-free nitrogen-doped carbon (denoted as N–C) was prepared for comparison. The scanning electron microscopy (SEM) (Supplementary Fig. 1) and transmission electron microscopy (TEM) (Fig. 1a) images of Co–N–C show uniform hollow spheres (mean diameter ca. 130 nm), which are similar to those of N–C. The high-resolution (HR)-TEM images do not show any sight of Co nanoclusters (Fig. 1b), implying that the cobalt species must be highly dispersed as tiny clusters and/or single atoms that are undetectable by or invisible to the HR-TEM technique. The aberration-corrected STEM images of Co–N–C (Fig. 1c, d) show individual bright dots. They represent the presence of Co atoms which are much heavier than the C and N atoms. The individual Co atoms uniformly dispersed throughout. Single Co atoms are repeatedly observed in different regions of Co–N–C (Supplementary Fig. 2a–c), and the statistical results of Co size in terms of ~250 particles (Supplementary Fig. 2c) show that the particle size of these Co species is ~1.3 Å (Supplementary Fig. 2d), further confirming that Co predominantly exists as single atoms rather than in the form of small clusters or nanoparticles. The HAADF-STEM mappings (Fig. 1e) confirm the existence of C, N and Co, as well as the homogeneous distribution of Co atoms throughout the catalyst. XRD patterns (Supplementary Fig. 3a) of Co–N–C and N–C are similar, and the two weak peaks at $2\theta$ of 22.8° and 44.0° can be attributed to the (002) plane of graphitic and (100) plane of disordered carbon[10,16], respectively (Supplementary Fig. 3). In comparsion with CoPc (Supplementary Fig. 3b), no peaks assignable to Co or CoPc species can be found in the XRD pattern of Co–N–C, in agreement with the fact that the Co entities are in sub-nanometric scale. Raman spectrum of Co–N–C (Supplementary Fig. 4) displays G and D bands at 1333 and 1596 cm$^{-1}$, respectively, corresponding to the characteristic of mesoporous carbonaceous flakes[17].

To probe the state of Co species at atomic level, ex situ X-ray absorption near-edge structure (XANES) and extended X-ray absorption fine structure (EXAFS) analyses were conducted. The pre-edge peak for cobalt phthalocyanine (CoPc) at 7716.5 eV is assigned to the forbidden $1s \rightarrow 3d$ transition. This peak originates from the Co−N$_4$ coordination that corresponds to a planar central symmetry structure (D$_{4h}$)[10,18,19]. Compared with the CoPc reference (Fig. 1f and Supplementary Fig. 5), Co–N–C is obviously lower in pre-edge peak intensity (feature b), indicating

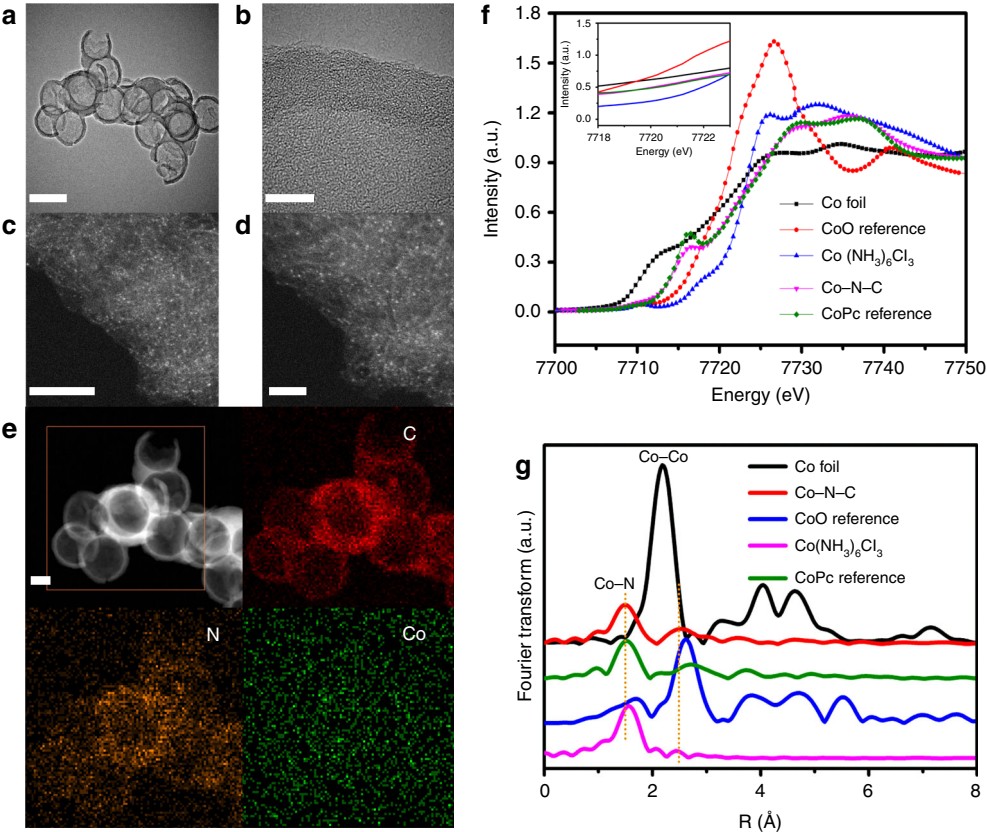

**Fig. 1 Physical characterization. a** TEM, scale bar: 200 nm; **b** HR-TEM, scale bar: 10 nm; and **c**, **d** Aberration-corrected STEM images; scale bar: 5 nm of **c** and 2 nm of **d**; **e** HAADF-STEM images of Co-N–C; STEM-mapping of N-K; C-K; Co-K, scale bar: 50 nm; **f** Normalized Co K-edge XANES spectra (inset is the magnified image of **f** in the range of 7718–7722 eV) and **g** Co K-edge EXAFS spectra of Co-N–C and references.

partial disruption of the planar central symmetry upon heat treatment[20,21]. The position of the absorption edge could be used as an indicator to estimate the valence states of Co species. According to the magnified XANES results (Fig. 1f), the absorption edge position of Co-N–C is located between that of CoO and $Co(NH_3)_6Cl_3$, suggesting that the single Co atom carries positive charge with valence state between +2 and +3.

The oxidation state of surface Co was confirmed by XPS analysis (Supplementary Fig. 6). Before $Ar^+$ etching, the binding energies of $Co2p_{3/2}$ peak in the cases of fresh CoPc and Co-N–C are 780.9 and 780.8 eV, respectively, which are higher than that of $Co^0$ (778.1 eV) and $Co^{2+}$ (779.2 eV), and only slightly lower than that of $Co^{3+}$ (781 eV)[10], suggesting that the dominant states of Co in both fresh samples are +3 (Supplementary Fig. 6a). After $Ar^+$ etching for 60 s under vacuum (Supplementary Fig. 6b), the binding energy of the $Co2p_{3/2}$ peak of CoPc is 779.05 eV, which is close to that of $Co^{2+}$ species (779.20 eV). As for Co-N–C, the binding energy of the $Co2p_{3/2}$ peak is 780.7 eV, suggesting that the dominant state of Co is +3. The Fourier-transformed (FT) $k^3$-weighted EXAFS spectrum of Co-N–C exhibits a distinct peak at 1.4 Å that matches well with the shell of Co-N (Fig. 1g)[18]. Since there is no detection of Co-Co coordination peak in comparison with Co foil reference (Fig. 1g), the absence of metallic clusters can be confirmed. Based on the above details, it is deduced that the single Co atoms of Co-N–C located in symmetric coordination and stabilized by N atoms are highly dispersed on the N-doped carbon hollow sphere.

Moreover, the Fourier transform EXAFS curves and the fitting results are shown in Supplementary Fig. 7 and Supplementary Table 2. The second weak peak at 2.3 Å is attributable to the replacement of the second shell of Co-N[18]. The CN over Co−N

is 5. Specifically, the peaks at 1.4 and 2.3 Å can be associated with Co-N CN of around 3.5 and 1.5 (Supplementary Table 2), respectively. According to the nitrogen adsorption–desorption isotherm of Supplementary Fig. 8, the Brunauer–Emmett–Teller (BET) surface area of Co-N–C is 356 $m^2 g^{-1}$, and the average pore diameter is ca. 7.64 nm (Supplementary Table 1).

**Ammonia synthesis performance.** Co-N–C and N–C were evaluated for $NH_3$ synthesis in a feed with composition of 25% $N_2$–75%$H_2$ at WHSV of 60,000 mL $g^{-1} h^{-1}$. The $NH_3$ synthesis rate over Co-N–C at 250–350 °C is much higher than that of N–C (Fig. 2a). Hence, it is reasonable to deduce that a proper assembly of single-atom Co center and nitrogen and/or carbon is required for effective $NH_3$ synthesis. To further probe the nature of the active sites in Co-N–C, nitrogen-free cobalt-doped carbon (marked as Co/C, the corresponding structure and textural properties are described in Supplementary Figs. 3 and 8, respectively) was synthesized with Co content of 3.80 wt% as confirmed by ICP-AES analysis (Supplementary Table 1). At 350 °C and 1 MPa, $NH_3$ synthesis rate over Co-N–C is 4.34 $mmol_{NH3} g_{cat}^{-1} h^{-1}$, which is 11-fold and 17-fold that of Co/C and N–C, respectively. The surface-area-normalized $NH_3$ synthesis rates were calculated and compared in Supplementary Fig. 9. A $NH_3$ synthesis rate of $3.36 \times 10^{-6}$ mmol $m^{-2} s^{-1}$ was obtained at 350 °C and 1 MPa on Co-N–C, which is 12-fold that of Co/C and 26-fold that of N–C, further confirming that the single Co sites in the form of Co-$N_x$ are responsible for the superior catalytic performance of Co-N–C.

In Fig. 2b, the $NH_3$ synthesis rates of selected Co-based catalysts at 350 °C and 1 MPa are compared, and Co-N–C is

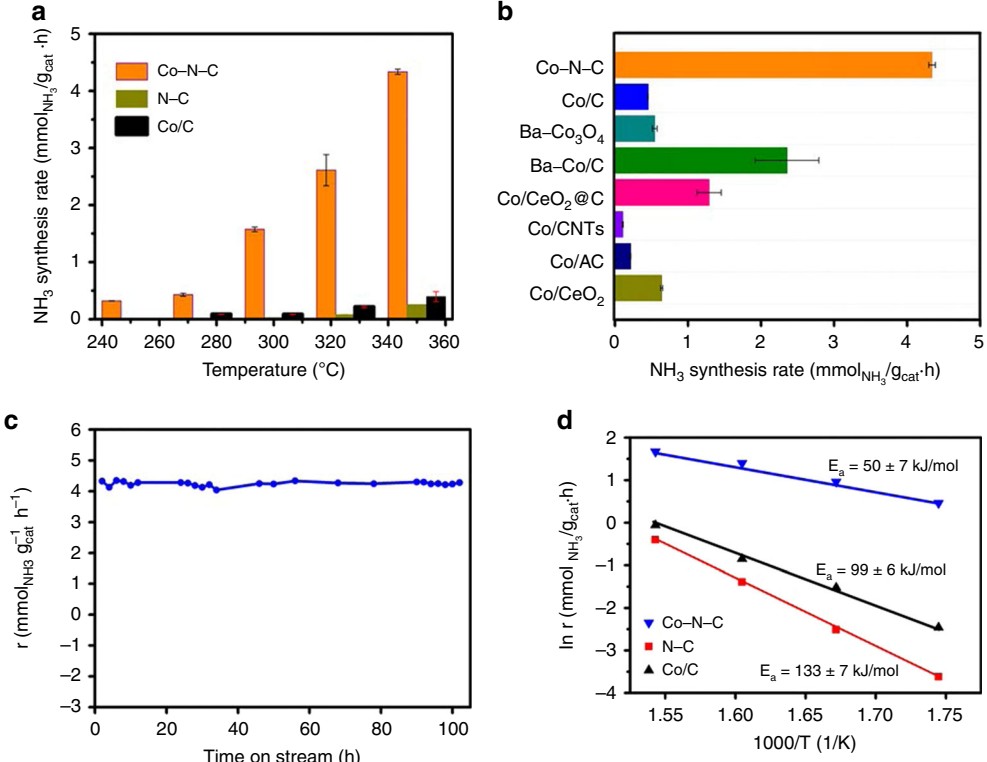

**Fig. 2 Catalytic performances. a** NH$_3$ synthesis rate for Co-N-C, Co/C, and N-C under 1 MPa and **b** NH$_3$ synthesis rate of selected catalysts at 350 °C under 1 MPa (The data points and error bars represent the average and standard deviation based on multiple measurements on the same catalyst at different times over different batches of samples). **c** Time course of NH$_3$ synthesis rate at 350 °C over Co-N-C and **d** Arrhenius plots.

much superior to the conventional Co catalysts. In terms of NH$_3$ synthesis rate per gram of cobalt, it is 116.35 mmol$_{NH3}$ g$_{Co}^{-1}$ h$^{-1}$ over Co-N-C at 350 °C, which is much superior to those of the traditional Co-based catalysts displayed in Supplementary Table 3. Additionally, NH$_3$ synthesis rate over Co-N-C exhibits an approximately linear increase with increasing pressure, from 2.92 to 12.91 mmol$_{NH3}$ g$_{cat}^{-1}$ h$^{-1}$ when the pressure was increased from 0.2 to 5.0 MPa at 350 °C (Supplementary Fig. 10). To evaluate the intrinsic activity of Co-N-C, we estimated the turnover frequency (TOF$_{Co}$), which indicates the activity of a catalyst on a per-Co-active-site basis. Surprisingly, TOF$_{Cototal}$ and TOF$_{Cosur}$ of Co-N-C reaches $1.91 \times 10^{-3}$ and $4.54 \times 10^{-3}$ s$^{-1}$ at 350 °C under 1 MPa (Supplementary Fig. 11), respectively.

The stability of the single-atom catalyst is a significant factor for practical use, and we tested the thermal stability of Co-N-C at 350 °C for 102 h and observed no obvious deactivation (Fig. 2c). During NH$_3$ synthesis at either 350 or 400 °C for 100 h under 1 MPa, the outlet CH$_4$ concentration is negligibly low (Supplementary Fig. 12), suggesting that under the adopted conditions the carbon support is highly stable. The structure of the used Co-N-C catalyst was examined by AC-STEM, and the images reveal atomic dispersion of Co and there is no obvious formation of nanoparticles or clusters (Supplementary Fig. 13). The excellent stability of the single-Co-atom sites in Co-N-C can be attributed to the coordination between the isolated Co atoms and adjacent N atoms. The XRD pattern (Supplementary Fig. 14) of used Co-N-C catalyst shows no significant change, indicating the absence of bulk transformation. Evidently, Co-N-C is a stable catalyst active for NH$_3$ synthesis under mild conditions. Kinetic analysis was performed over Co-N-C to clarify the reaction mechanism. The apparent activation energy ($E_a$) obtained in the 300–375 °C range at 1 MPa is 50 ± 7 kJ/mol (Fig. 2d), which is similar to those of Co-LiH (40–60 kJ/mol)[22] and Co/C12A7:e$^-$

(~50 kJ/mol)[23]. The researchers reported that the dissociation of N≡N triple bonds is not a rate-limiting step over Co-LiH and Co/C12A7:e$^-$, owing to the much lower $E_a$ values in comparison with those of traditional Co-based catalysts (Supplementary Table 3). To be noted, in Ar-TPD-MS investigation conducted over Co-N-C after NH$_3$ synthesis (Supplementary Fig. 15), the primary desorption species are *N$_2$H and *N$_2$H$_2$ rather than *NH or *NH$_2$, implying that N$_2$ is preferentially hydrogenated without undergoing direct dissociation on the single-atom Co-N-C catalyst, as demonstrated by DFT calculation. Accordingly, the $E_a$ value over Co-N-C is not noticeably high, indicating that N$_2$ hydrogenation can take place under mild conditions.

To find out whether the N species of Co-N-C could be involved in NH$_3$ synthesis, we exposed Co-N-C to 10%H$_2$/Ar at 350 °C and 1 MPa, and the cumulative amount of NH$_3$ as a function of time is provided in Supplementary Fig. 16. Obviously, there is the formation of NH$_3$ which comes to a steady state after 6 h. The amount of NH$_3$ formed is 1.12 mmol$_{NH3}$ per gram of catalyst, equivalent to the consumption of 29.3% of the nitrogen in Co-N-C. This observation confirms that there are N species in Co-N-C that can be hydrogenated to NH$_3$ plausibly by a process of chemical looping.

**In situ XANES and EXAFS.** The intrinsic mechanism of NH$_3$ synthesis could be disclosed by cutting edge operando techniques. Using a home-built cell (Supplementary Fig. 17), we performed operando XANES (Supplementary Fig. 18) and EXAFS (Supplementary Figs. 19 and 20) measurements to unveil the nature of the single atom Co active sites during NH$_3$ synthesis. With the proceed of reaction from 0 to 60 min in the presence of 10%H$_2$/He (Supplementary Fig. 18a) or N$_2$-H$_2$ mixture (V$_{N2}$:V$_{H2}$ = 1:3, Supplementary Fig. 18b), the Co K-edge XANES profiles are

almost the same. Moreover, the Co K-edge EXAFS curve fitting of Co–N–C demonstrates the lack of Co–Co coordination in either 10%$H_2$/He (Supplementary Fig. 19a–d) or $N_2$–$H_2$ mixture (Supplementary Fig. 20) atmosphere, further confirming the absence of metallic clusters during $NH_3$ synthesis. The observation is consistent with the lack of change in the HAADF-STEM images and XRD patterns of the fresh and used Co–N–C samples. Moreover, the CN value of the first Co–N shell in Co–N–C is still 3.5 with the proceed of reaction in the presence of either 10%$H_2$/He or $N_2$–$H_2$ mixture (Supplementary Table 2), further demonstrating that the single atom $Co_1$–$N_{3.5}$ active sites can remain stable in $NH_3$ synthesis atmosphere. To be noted, the CN of second Co–N shell varies in the range of 0.9–1.5 upon exposure to 10%$H_2$/He. Because the fitting error by itself was about 20%, it is meaningless to make any deduction based on the change of CN. Overall, the results suggest that the CN remains stable under the $NH_3$ synthesis atmosphere.

**$H_2$ pulse and isothermal isotope-labeling experiments**. In order to explore the involvement of N species during $NH_3$ production, $H_2$-pulsing experiment and isothermal surface reaction (ISR) were performed at 350 °C in the adopted temperature range of $NH_3$ synthesis (see "Methods" section for details). During the 40 pulses of $H_2$ (Fig. 3a), the signal of $NH_3$ formation synchronized that of $H_2$ introduction, suggesting that the hydrogenation of the N species in Co–N–C is facile. In the ISR experiment in a feed of 10%$H_2$/Ar (Fig. 3b), the $NH_3$ signal decreases with time as a result of N consumption. With the purging of the Co–N–C sample by Ar for 4 h and subsequent treatment in $N_2$ atmosphere for 2 h, $NH_3$ signal can be almost restored to the previous value upon the feeding of 10%$H_2$/Ar. On the basis of the above results, we conclude that a Co–N–C sample exhausted in N species can be replenished by gas-phase $N_2$, it is confirmed that the weak-coordination of N at the atomically dispersed Co sites is essential for $NH_3$ production by means of chemical looping using $H_2$. To find out whether the $N_2$ from the gaseous phase used for nitrogen replenishment is involved in the subsequent formation of $NH_3$, a similar ISR was performed using isotopic $^{15}N_2$ (99% $^{15}N_2$, 30 min) for the replenishment (Fig. 3c). Upon the introduction of 10%$H_2$/Ar, there is the detection of $^{15}NH_3$ ($H_2O$ signal was deducted from $m/z = 18$, for more details, see "Methods" section), unequivocally confirming that the replenished N is entirely from $^{15}N_2$.

**$^{15}N_2$ isotopic exchange and in situ XPS experiments**. To monitor the transformation of nitrogen in Co–N–C, isothermal $^{15}N_2$ isotopic exchange (INIE) investigation was conducted at 350 °C. As shown in Fig. 4a, upon the introduction of $^{15}N^{15}N$, there is the detection of $^{15}N^{14}N$ and $^{14}N^{14}N$, with the former constantly higher than the latter, undoubtedly confirming that there are nitrogen atoms in Co–N–C that are exchangeable. It should be emphasized that the exchanged N atoms in Co–N–C after 30 min at 350 °C is ca. 30% (more details about the number of N atoms exchanged ($N_e$) in the case of Co–N–C are provided in Supplementary Information), which is close to the amount of N atoms involved in $NH_3$ production via chemical looping (ca. 29.3%, Supplementary Fig. 16). The exchange could be either homomolecular ($^{14}N_2(g) + ^{15}N_2(g) \leftrightarrow 2^{14}N^{15}N(g)$) or heterolytic ($^{15}N_2(g) + ^{14}N(s) \rightarrow ^{14}N^{15}N(g) + ^{15}N(s))$[24]. That the percentage of the $^{15}N$ gas-phase atomic fraction remains constant with the advance of reaction is indicative of the homomolecular route. The facile $^{15}N^{14}N$ production reveals that the isotopically exchangeable N species in Co–N–C are active, and can be hydrogenated to $NH_3$ via chemical looping. It is envisioned that the coordination between this kind of N species and the atomically dispersed Co is

not too strong so that they can undergo isotopic exchange with gas-phase $^{15}N_2$.

In the isothermal $^{15}N_2$ isotopic-labeling experiment (INILE), the Co–N–C catalyst was exposed to a mixture of $^{15}N_2$ and $H_2$ at 350 °C, and the results are illustrated in Fig. 4b, c. There is the detection of $^{14}N^{14}N$ ($m/z = 28$) and $^{14}N^{15}N$ ($m/z = 29$) (Fig. 4b) as well as $^{15}NH_3$ (Fig. 4c, $m/z = 18$, $H_2O$ signal was deducted from $m/z = 18$; for more details, see "Methods" section), and their amount decrease with time, in line with the formation of $^{14}N^{15}N$ (i.e., $^{14}N_2(g) + ^{15}N_2(g) \leftrightarrow 2^{14}N^{15}N(g)$) and $^{15}NH_3$ (i.e., $^{15}N_2 + 3H_2 \leftrightarrow 2^{15}NH_3$). The $m/z = 16$ signal could be due to $^{14}NH_2$ or $^{15}NH$, and because its intensity is almost constant when there is a decrease of $^{15}N_2$, it is reasonable to assign it to $^{14}NH_2$. As for the $m/z = 17$ signal, its intensity changes with the change of $^{15}N_2$ concentration (Supplementary Fig. 21); it is hence reasonable to assign it to $^{15}NH_2$. Similar to the results of ISR experiment, the formation of $^{14}NH_3$ is a result of chemical looping reaction involving the consumption of a certain amount of N species in Co–N–C. It is noted that the $m/z = 29$ ($^{14}N^{15}N$) signal is much larger than the $m/z = 17$ ($^{14}NH_3$) or $m/z = 18$ ($^{15}NH_3$) signal. Referring to these results and those of INIE investigation, if one assumes that $N_2$ dissociation is the rate-determining step, the resulted N species should immediately react with H to produce $NH_3$ rather than undergo isotopic exchange to produce a high amount of $^{14}N^{15}N$. It is hence deduced that $N_2$ dissociation is no more the bottleneck of $NH_3$ synthesis, but the formation of $^*N_2H_2$ bond is possibly the rate-determining step, as demonstrated by DFT calculation.

We chose in situ XPS analysis as a complimentary technique to acquire surface information to determine what kind of surface N species over Co–N–C would take part in $NH_3$ production via the chemical looping pathway. Figure 4d shows the N1$s$ spectrum of fresh Co–N–C and those of the catalyst exposed to 10%$H_2$/Ar and 25%$N_2$−75%$H_2$. For the fresh Co–N–C sample, four N species can be identified (detailed parameters provided in Supplementary Table 4). They are graphitic (400.7 eV), pyrrolic (399.4 eV), pyridinic (398.9 eV), and N-oxide (404.3 eV)[25]. The surface composition of pyridinic N species is 38.9% (Supplementary Table 4). The N1$s$ spectrum recorded after exposure to 10%$H_2$/Ar at 350 °C for 2 h shows a surface pyridinic N composition of 24.4%. In the case of exposing the catalyst to 25%$N_2$−75%$H_2$ at 350 °C for 2 h, the surface composition of pyridinic N species returns back to 38.7%. These results indicate that surface pyridinic N can be consumed and then regenerated under synthetic $NH_3$ atmosphere.

According to DFT calculation, the results of coordination affinity (Supplementary Fig. 22) illustrate that pyrrolic N and pyridinic N both can anchor single Co atoms. Interestingly, the bond length of pyridinic N interaction with Co ($d_{Co-N} = 3.36$ Å) is much larger than that of pyrrolic N interaction with single atom Co ($d_{Co-N} = 2.14$ Å, Supplementary Table 5), indicating pyridinic N interaction with Co is much weaker than the latter. The bond length of H–H is stretched from 0.75 Å of gas-phase $H_2$ to 2.31 Å upon adsorption on the surface pyridinic N species (Supplementary Table 5), and considering that $H_2$ adsorption energies on graphitic N, pyrrolic N and pyridinic N are 0.84, 0.43, and −1.98 eV, respectively (Supplementary Table 5), it is deduced that surface pyridinic N species could be easily activated by gas-phase $H_2$. The adsorbed $H_2$ spontaneously dissociated into two hydrogen atoms, one bonded to pyrrolic N and the other coordinated with the Co atom (Supplementary Fig. 23), and the overall process is exothermic (−2.87 eV). Based on the results of $^{15}N_2$ isotopic labeling and in situ XPS experiments as well as those of DFT calculation, it is clear that the pyridinic N interacting weakly with a Co atom could participate in the dynamic cyclic reaction. As confirmed by isothermal $^{15}N_2$

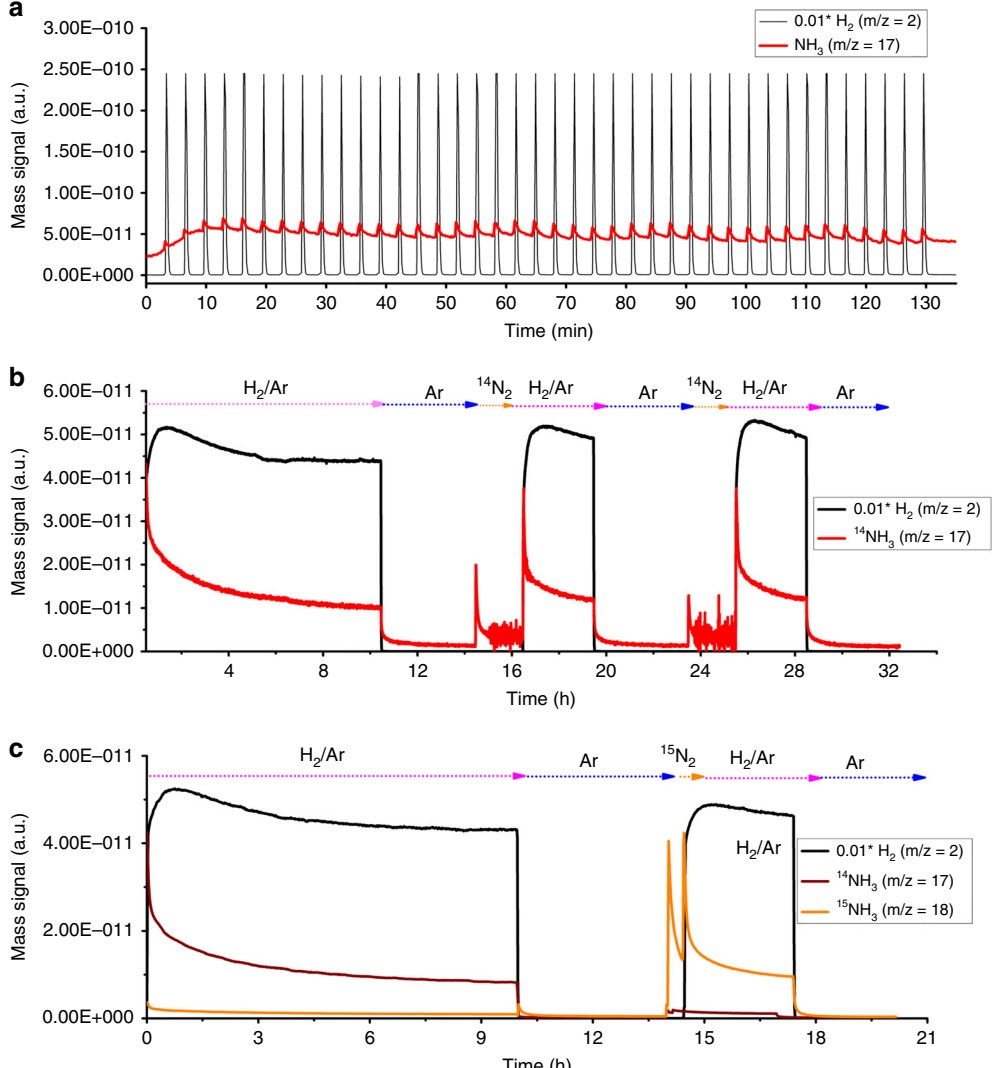

**Fig. 3 Identification of N-involved chemical looping pathway. a** $H_2$ pulse results at 350 °C, **b** isothermal surface reaction profiles; and **c** isothermal $^{15}N$ isotopic labeling results over Co–N–C at 350 °C.

isotopic exchange, ca. 30% of the N atoms in Co–N–C are exchangeable, and therefore it can be inferred that $x$ of the above dynamic Co–$N_x$ site is in the range of "$0 < x \leq 1.5$" during $NH_3$ synthesis process. Furthermore, according to the AC-STEM and in situ EXAFS results, it is deduced that the pyrrolic N serves to anchor a single Co atom, and the stabilized Co sites in the form of $Co_1$–$N_{3.5}$ remains active in the process of $NH_3$ synthesis.

**DFT calculations**. The production of $NH_3$ can follow either the dissociative (L–H mechanism) or associative route (E–R mechanism)[4,6,13,26]. DFT calculation was performed to understand the pathway of $NH_3$ synthesis over Co–N–C. The steps to construct the model of Co–N–C for $NH_3$ synthesis is shown in Supplementary Fig. 24 (please see Supporting Information for more details), and it can be observed that the Co atom is located at the center of phthalocyanine and coordinates with the N-dopant site on the carbon sphere. The changes of free energy for the formation of $NH_3$ and the detailed reaction steps of $N_2 + 3H_2 \rightarrow 2NH_3$ reaction on Co–N–C are illustrated in Fig. 5a and Supplementary Fig. 25, respectively. Notably, it is very difficult to dissociate $N_2$ directly on a single Co atom, while the hydrogenation of $^*N_2$ to $^{**}NHNH$ on single Co sites in the form of

steady-state $Co_1$–$N_{3.5}$ is feasible (Fig. 5a). The theoretical outcomes are in agreement with the Ar–TPD–MS result that $N_2$ molecules are first adsorbed on single Co sites in the form of $Co_1$–$N_{3.5}$, and then directly activated by hydrogen species to form $^*NHNH$ intermediates rather than undergoing direct dissociation to N atoms. In the present study, the $\Delta G$ values for the hydrogenation of $^*NHNH$ to $^*NH$–$NH_3$ and that of $^*NH$–$NH_3$ to $^*NH_2$–$NH_4$ on single atom $Co_1$–$N_{3.5}$ sites is −2.608 and −2.411 eV, respectively. Finally, $^*NH_2$–$NH_4$ can transform into two $NH_3$ molecules with $\Delta G$ value of −0.769 eV (Fig. 5a). The $NH_3$ synthesis pathway over the single Co sites in the form of steady-state $Co_1$–$N_{3.5}$ is illustrated in Fig. 5b. In the absence of $B_5$ sites, the single atom $Co_1$–$N_{3.5}$ sites enable $N_2$ adsorption and hydrogenation, and then $NH_3$ is much easily generated via the breaking of $^*NH_2$–$NH_4$ bond. Following the E–R mechanism in such a way, the bottleneck of direct N≡N dissociation is bypassed and $NH_3$ synthesis is made possible at mild conditions.

On the basis of the results of $H_2$ pulsing experiment, $^{15}N_2$ isotopic labeling and in situ XPS investigations, together with those of DFT calculation, we propose a dynamic cyclic reaction pathway for $NH_3$ production via chemical looping. As shown in Fig. 5c, the pyridine N weakly coordinated to a single Co atom reacts with adsorbed $H_2$ to produce $NH_3$ and simultaneously

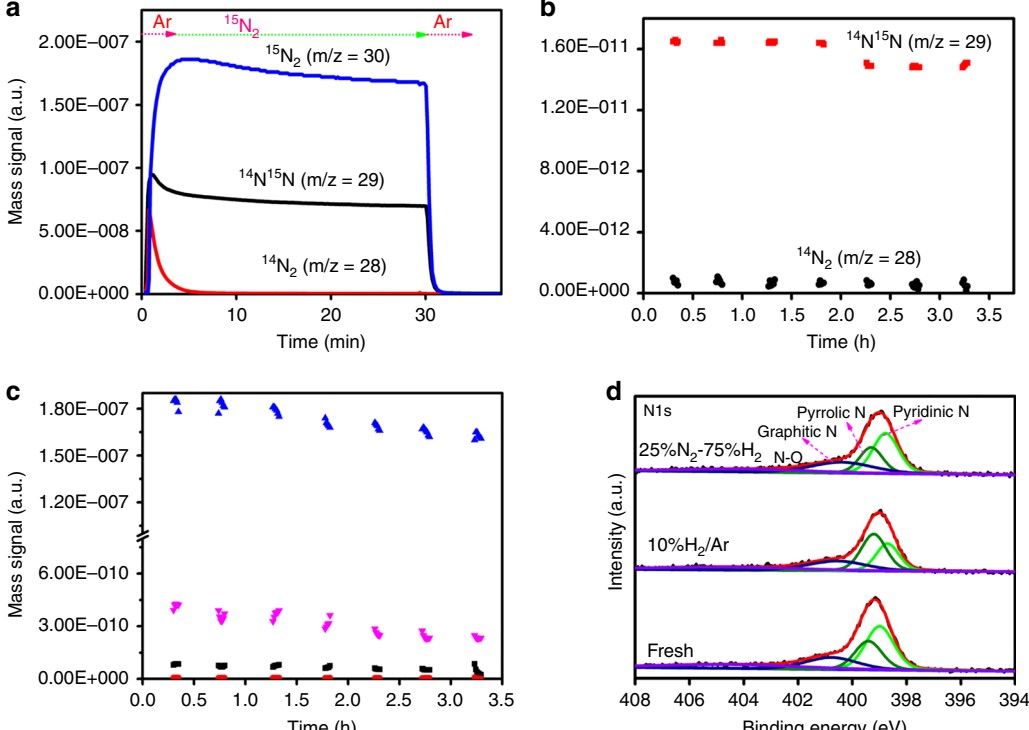

**Fig. 4 Isotopic labeling and in situ XPS studies.** Isotopomer distribution of **a** $^{15}N_2$ isotopic exchange experiment at 350 °C in the feeding of $^{15}N_2$, **b** $^{15}N_2$ isotopic exchange experiment at 350 °C in the feeding of $^{15}N_2$ and $H_2$ over Co-N-C at 350 °C, **c** $^{15}N_2$ isotopic-labeling experiment over Co-N-C at 350 °C in the presence of $^{15}N_2$ and $H_2$ (blue dots are the signal of $m/z = 30$ ($^{15}N_2$); magenta dots are the signal of $m/z = 18$ ($^{15}NH_3$); black dots are the signal of $m/z = 17$ ($^{15}NH_2$) and red dots are the signal of $m/z = 16$ ($^{14}NH_2$), **d** XPS spectra of N1s over Co-N-C under different exposure atmospheres.

leaving behind an anionic N vacancy. The N atoms in Co–N–C are exchangeable and can be replenished by gaseous $N_2$ upon removal. The cycle of replenishment is (i) $N_2$ adsorption on anionic nitrogen vacancies, and (ii) $N_2$ from occupied $V_N^*$ are fixed to bond with Co sites by donating electrons to the unoccupied $d$ orbitals of Co[1,27], hence restoring the initial state of Co–N–C. The dynamic cyclic sites of Co–N–C and the steady-state reaction of $Co_1–N_{3.5}$ are routes for energy-efficient synthesis of $NH_3$ under mild conditions.

## Discussion

To summarize, we have demonstrated the atomically dispersed Co–$N_x$ sites are responsible for the outstanding performance of Co–N–C in $NH_3$ synthesis under mild conditions, displaying an exceptionally high $NH_3$ synthesis rate of 116.35 $mmol_{NH3}$ $g_{Co}^{-1}$ $h^{-1}$. The results of experimental investigations and theoretical calculations reveal that the excellent activity can be related to the single Co sites in the form of steady-state $Co_1–N_{3.5}$ and single Co sites in the form of dynamic Co–$N_x$ ($0 < x ≤ 1.5$). The former enable $N_2$ adsorption and hydrogenation as well as the subsequent formation of $NH_3$ via the breaking of $*NH_2–NH_4$ bond following the E–R mechanism, while the latter afford surface pyridinic N to anchor single Co atoms for $NH_3$ production via chemical looping. The results demonstrate that the dual active sites release $NH_3$ synthesis from the bottleneck of N≡N dissociation, leading to superior $NH_3$ production under mild conditions. It is anticipated that such understandings on Co–N–C shed light on the design of SAC with multiple active sites for efficient $NH_3$ synthesis.

## Methods

**Chemicals and materials**. N,N-Dimethylformamide (DMF, AR) and melamine (99%) were purchased from Shanghai Macklin Biochemical Co., Ltd. Resorcinol

(99%) and formaldehyde as well as tetraethoxysilane (TEOS, 99.99%) were purchased from Aladdin. CoPc (97%) and F127 ($PEO_{106}PPO_{70}PEO_{106}$) as well as cobalt chloride hexahydrate (99.99%) were purchased from Sigma-aldrich. Ammonia aqueous solution (25–28 wt%) was purchased from Xilong Scientific Co., Ltd. High purity helium (99.9999%) and argon (99.9999%) as well as nitrogen (99.9999%) gases were purchased from Linde Industrial Gases; the $N_2$–$H_2$ and $H_2$–Ar mixture gases of designated proportions were also purchased from Linde Industrial Gases. $^{15}N_2$ (98%) was supplied by Cambridge Isotope Laboratories, Inc. High purity 10%$H_2$–90%He (99.9999%) mixture gas used for in situ XAS experiment was supplied by Beiwen Gas Manufacturing Plant in Beijing.

**Catalyst preparation**. Synthesis of hollow carbon and N-doped hollow Carbon spheres (CSs): Taking N-doped hollow CSs as an example, 10 mL of $NH_3$ solution (25–28 wt%), 240 mL of ethanol, and 80 mL of deionized water were mixed, and stirred at room temperature (RT) for 1 h. Subsequently, 11.2 mL of tetraethyl orthosilicate was added to the above solution and the resulted mixture was stirred for 1 h. Then 1.2 g of F127, 1.6 g of resorcinol, and 2.24 mL of formaldehyde were added and the resulted solution was stirred for 0.5 h. Thereafter, 1.26 g of melamine and 1.68 mL of formaldehyde were sequentially added, and the mixture was stirred and heated at 100 °C for 24 h. The solid product was collected by filtration and washed with deionized water, and then slowly heated (2 °C min$^{-1}$) in a quartz-tube furnace (OTF-1200X) to 700 °C under an Ar flow of 50 mL/min, and calcined at 700 °C for 2 h. Finally, $SiO_2$ was removed by etching with HF solution, and the resulted substance was collected by filtration, washed with deionized water, and dried at 120 °C for 12 h. The as-prepared N-doped carbon support is herein referred to as N–C. Additionally, the synthesis of CSs is similar to that of N–C but without the introduction of nitrogen source (see Supplementary Methods).

Synthesis of Co–N–C: Typically, 60 mg of N–C and 40 mg of CoPc were added to 60 mL of DMF and the mixture was stirred for 2 h. Then, a CoPc–DMF solution was added to the N–C suspension, and the resulted mixture was stirred for 24 h. Finally, the product was dried in a vacuum oven at 60 °C for 12 h and heated to 500 °C at a rate of 2 °C min$^{-1}$ under Ar flow for 2 h. For comparison, 3.8 wt%Co/C was also synthesized via incipient wetness impregnation using $CoCl_2·6H_2O$ as cobalt precursor.

The preparations of other Co-based catalysts (Co/C, Ba–$Co_3O_4$, Co/CeO$_2$@C, Co/CNTs, Co/AC, and Co/CeO$_2$) for comparison purposes are detailed in Supplementary Information.

Synthesis of Co/CeO$_2$: The preparation of Co/CeO$_2$ was by simple co-precipitation. Nitrate salts of cobalt and cerium were simultaneously dissolved in deionized water. The content of Co was 4 wt% against CeO$_2$. Aqueous ammonia was then added dropwise into the mixture with vigorous stirring. The resultant

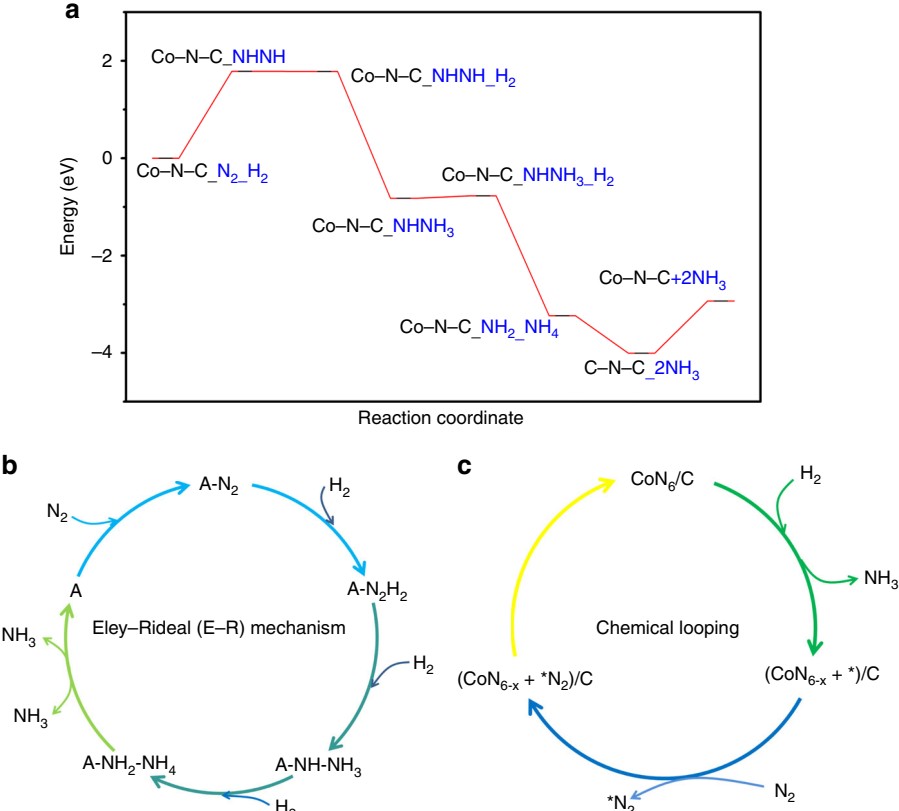

**Fig. 5 Theoretical investigation and reaction pathway. a** Changes of free energy for the formation of $NH_3$ on Co–N–C; **b** $NH_3$ synthesis pathway on single Co sites in the form of steady-state $Co_1$–$N_{3.5}$ (A represents the active sites); **c** $NH_3$ production on dynamic cyclic sites via chemical-looping pathway ($x$ is in the range of $0 < x \leq 1.5$ and $V_N^*$ stands for an anionic nitrogen vacancy; herein $CoN_6$/C only represents the molar ratio of each element in the Co–N–C catalyst).

slurry was stirred for 0.5 h and washed several times with deionized water, then aged at 40 °C for 4 h and dried at 120 °C for 24 h to give the Co/CeO₂ precursor, which was subject to heat treatment at 450 °C for 4 h.

Synthesis of Co/CeO₂@C: The synthesis process was based on a previously reported method[28]. Typically, 20 g of glucose monohydrate was dissolved in 180 mL of deionized water, and a colorless solution was obtained after 15 min of vigorous stirring. Then the solution was transferred to and sealed in a stainless steel autoclave and kept at 180 °C for 9 h to afford a dark brown solid product, which was collected and dried at 120 °C for 24 h. The obtained carbon-sphere templates are herein labeled as CSs.

Next, 0.2 g of CSs was dispersed in 30 mL of absolute ethyl alcohol and subject to sonication of 0.5 h. Then 50 mL of a solution consisting 0.1302 g of Ce (NO₃)₃·6H₂O and 0.13 g of hexamethylenetetramine (HMT) was poured into the CSs slurry. The resulted mixture was stirred for 2 h at RT and subject to reflux at 75 °C for 4 h to give CeO₂@CSs nanospheres which were collected by suction filtration. The CeO₂@CSs precursor black in color was dried at 120 °C for 24 h.

Finally, 0.2 g of CeO₂@CSs was dispersed in 30 mL of absolute ethyl alcohol and subject to sonication of 0.5 h to give a slurry. Then 50 mL of a solution composed of nitrate salts of cobalt (4 wt% Co against CeO₂@CSs) and 0.13 g of HMT was added to the slurry, and the mixture was stirred for 2 h at RT, followed by reflux at 75 °C for 4 h. The as-produced substance was collected and dried at 70 °C for 6 h, and calcined at 450 °C for 4 h under Ar atmosphere. The obtained material is herein denoted as Co/CeO₂@C.

Synthesis of Ba/Co₃O₄: Typically, 0.028 g of barium acetate (C₄H₆BaO₄) was dissolved in 1 mL of deionized water, and added to a mixed solution of cobalt nitrate and dimethylimidazole, and the as-resulted mixture was subject to agitation at RT for 3 h. After centrifugation, washing, and drying, a purple product was collected. Then the obtained product was impregnated with a certain amount of methanol solution in which furfuryl alcohol (FA) was dissolved. The FA in the pores underwent polymerization when heated in a muffle furnace at 80 and 150 °C for 14 and 6 h (heating rate = 2 °C/min), respectively. Subsequently, carbonization was carried out at 450 °C for 4 h (heating rate of 2 °C/min), and the obtained product is donoted herein as Ba–Co₃O₄.

Synthesis of Co/CNTs and Co/AC: Co/CNTs and Co/AC were prepared by impregnation method. The procedures for the generation of activated carbon (AC) were reported elsewhere[29]. The CNTs was purchased from nanoscience and

technology companies in Nanjing of China, and Co was loaded on CNTs by impregnation using aqueous nitrate salt of cobalt (4 wt% Co against CNTs). The as-resulted mixture was dried at 120 °C overnight, and calcined at 450 °C for 4 h. The obtained sample is herein denoted as Co/CNTs. The Co/AC sample was prepared following the same procedures but with CNTs replaced by AC.

Physical characterization: Aberration-corrected high-angle annular dark-filed scanning transmission electron microscopy (HAADF-STEM) was conducted on a JEOL JEM-ARM 200 F instrument equipped with a CEOS probe corrector, with a guaranteed resolution of 0.08 nm. SEM was performed on a Hitachi Model S-4800 microscope operated at 5 kV. TEM and HR-TEM were conducted on a JEM-2010 microscope. High-angle annular dark field scanning transmission electron microscopy (HAADF-STEM) mapping was performed over a JEOL JEM-ARM 200 F instrument. Powder X-ray diffraction (XRD) was performed on a Panalytical X'Pert Pro diffractometer using Cu-K$_\alpha$ radiation ($\lambda = 0.1789$ nm) operating at 40 kV and 40 mA. The BET surface area and Barrett–Joyner–Halenda (BJH) pore size distribution were measured by $N_2$ adsorption–desorption on a Micromeritics ASAP 2020 instrument at −196 °C after the sample was degassed at 120 °C for 2 h in vacuum. Elemental analysis (EA) was performed on a Vario EL-Cube. The Raman measurement was performed on a multi-channel modular triple Raman system (Renishaw Co.) with confocal microscopy at RT excited with the 532 nm line of an Ar laser. A 50 microscope objective lens was applied for focusing the laser beam and collection of the scattered light. The spot diameter of the focused laser beam on the sample was about 1 mm and a typical spectrum acquisition time was 50 s. ICP-AES analysis was conducted using an Ultima 2 spectrometer.

Evaluation of performance for ammonia synthesis: Before the evaluation of catalytic performance for $NH_3$ synthesis, the samples (0.25 g, diluted with quartz powder in a 1:8 volumetric ratio) were reduced in a flow of 10%$H_2$/Ar at 400 °C for 4 h. Under the condition for $NH_3$ synthesis in a 25%$N_2$–75%$H_2$ mixture at a WHSV of 60,000 mL g$^{-1}$ h$^{-1}$ and a given pressure, the outlet $NH_3$ concentrations were measured using a known amount of diluted $H_2SO_4$ solution (0.02 mol L$^{-1}$), as well as analyzed by ion chromatography (Thermo Scientific, DIONEX, ICS-600). Finally, the $NH_3$ synthesis rates were acquired on the basis of outlet $NH_3$ concentrations.

Turnover frequency (TOF$_{Co total}$) was acquired having the $NH_3$ synthesis rate divided by total number of Co atoms, while TOF$_{Co sur.}$ was obtained having the $NH_3$ synthesis rate divided by surface number of Co atoms.

Available nitrogen testing: A sample of 0.25 g of Co–N–C diluted with quartz powder was exposed to a $10\%H_2/Ar$ mixture at 350 °C and 1 MPa with WHSV of 60,000 mL g$^{-1}$ h$^{-1}$; the outlet $NH_3$ concentration was analyzed by ion chromatography (Thermo Scientific, DIONEX, ICS-600).

**Experiments for methanation determination.** In the determination of the possibility of methanation, 0.25 g of Co–N–C was exposed to a flow of $10\%H_2/Ar$ at 400 °C for 4 h. In another case, the conditions were under a $25\%N_2$–$75\%H_2$ mixture at WHSV of 60,000 mL g$^{-1}$ h$^{-1}$ and pressure of 1 MPa. In both experiments, the outlet $CH_4$ concentrations were measured using an on-line GC–Mass (GCMS-QP2010 SE).

Temperature-programmed Argon (Ar) desorption: Ar-TPD-MS experiment was performed by mass spectrometry using an Autochem 2920 instrument. After the $NH_3$ synthesis activity test, 50 mg of the catalyst was recovered and flushed with Ar before being heated to 600 °C at a rate of 10 °C min$^{-1}$. The desorbed species were monitored by recording the signals at $m/z = 30$ ($N_2H_2$), 29 ($N_2H$), 17 ($NH_3$), 16 ($NH_2$), and 15 (NH).

**Isothermal $^{15}N_2$ isotopic-labeling experiment.** INILE was performed by mass spectrometry employing an Autochem 2920 instrument. Before the experiment, the carrier gas pipeline was purged with $^{15}N_2$ (30 mL/min) for 15 min. Then, a sample of 50 mg was heated from RT to 350 °C in Ar, and then the catalyst was exposed to a $10\%H_2/Ar$ atmosphere at 350 °C for 10 h, and flushed with Ar for 4 h. Next, the sample was exposed to $^{15}N_2$ for 30 min. A short time gas flow system was adopted to minimize the use of expensive $^{15}N_2$. Then, the feed gas was once again changed to $10\%H_2/Ar$ at 350 °C and followed by Ar flushing. The $m/z = 2$, $m/z = 17$, and $m/z = 18$ signals were recorded as a function of time. It should be emphasize that the $m/z = 18$ signal of $H_2O$ over $CoN_6/C$ was also collected under similar experimental condition but with $^{14}N_2$ rather than $^{15}N_2$ (99% $^{15}N_2$) being fed in, and the $H_2O$ signal was recorded to be deducted from the $m/z = 18$ signal of the $^{15}N_2$ experiment to obtain the genuine $^{15}NH_3$ signal.

Isothermal $^{15}N$ isotopic exchange experiment: $^{14}N$–$^{15}N$ isotope exchange experiment was performed also using the Autochem 2920 instrument. A Co–N–C sample of 50 mg was heated from RT to 350 °C in Ar, and then the catalyst was exposed to $^{15}N_2$ at 350 °C for 30 min. The signals of $N_2$ ($m/z = 28$), $^{14}N^{15}N$ ($m/z = 29$), and $^{15}N_2$ ($m/z = 30$) were measured as a function of time.

Isothermal surface reaction: ISR experiment was performed by mass spectrometry employing an Autochem 2920 instrument. A Co–N–C sample of 50 mg was heated from RT to 350 °C in an Ar atmosphere, and then the catalyst was exposed to a $10\%H_2/Ar$ atmosphere at 350 °C for 10 h, and then flushed with Ar for 4 h. Next, the sample was exposed to $N_2$ for 2 h, and then the feed gas was changed to $10\%H_2/Ar$ at 350 °C and followed by Ar flushing. The $m/z = 2$ ($H_2$), $m/z = 17$ ($NH_3$), and $m/z = 18$ ($H_2O$) signals were monitored as a function of time. Additionally, a similar ISR experiment was performed on N–C but with the feed time of $10\%H_2/Ar$ and $N_2$ shortened.

XANES and EXAFS measurements: The ex situ XANES and EXAFS analyses were performed at the 1W2B beam line of Beijing Synchrotron Radiation Facility. The Co K-edge spectra of samples and references in fluorescence mode were measured at RT. A Si(111) double-crystal monochromator was used to abate the harmonic content of the monochrome beam.

In situ XAS measurments: In situ XAS was also performed at the 1W2B beam line of Beijing Synchrotron Radiation Facility. The apparatus for the experiment was provided by Beijing Institute of Chemical Industry of Sinopec, which contains an in situ cell and systems for gas circulation and sample heating (Supplementary Fig. 17). Catalyst sample was cast pressed into uniform flakes to allow fluorescence signals to pass from the sample to the detector. Then in situ XAS data was collected at 350 °C in the presence of $10\%H_2/He$ or $N_2$–$H_2$ ($V_{N2}:V_{H2}=1:3$) as a function of time.

EXAFS data analysis: The acquired EXAFS data was treated on the basis of the standard procedures using the ATHENA module implemented in the IFEFFIT software packages. The $k^3$-weighted EXAFS spectra were gained by subtracting the post-edge background from the overall absorption and then normalized with respect to the edge-jump step. Subsequently, the $k^3$-weighted $\chi(k)$ data of Co K-edge was Fourier transformed to real (R) space employing a Hanning windows (dk = 1.0 Å$^{-1}$) to segregate the EXAFS contributions from different coordination shells. To acquire the quantitative structural parameters around central atoms, least-squares curve parameter fitting was implemented via adopting the ARTEMIS module of IFEFFIT software packages[30].

Hydrogen pulse experiment: $H_2$ pulse experiment was conducted on an Autochem 2920 instrument. A catalyst sample of ca. 50 mg was heated from RT to 350 °C in an Ar atmosphere. Afterward, using Ar as carrier gas (flow rate = 30 mL/min), $10\%H_2/Ar$ was pulsed in at 350 °C for 40 times. The mass signals of $m/z = 2$ and $m/z = 17$ were measured as a function of $H_2$ pulses.

Isothermal $^{15}N_2$ and hydrogen reaction: Isothermal $^{15}N_2$ and $H_2$ reaction was performed using an Autochem 2920 instrument. A sample of 50 mg was heated from RT to 350 °C in an Ar atmosphere, and then the catalyst was exposed to $^{15}N_2$ and $H_2$ in volume ratio of 1:3 at 350 °C for 30 min. The signals of $N_2$ ($m/z = 28$), $^{14}N^{15}N$ ($m/z = 29$), $^{15}N_2$ ($m/z = 30$), and $m/z = 17$ ($^{14}NH_3$, $^{15}NH_2$), $m/z = 16$ ($^{14}NH_2$, $^{15}NH$) as well as $m/z = 18$ ($^{15}NH_3$) were measured as a function of time. It should be emphasize that the $m/z = 18$ signal of $H_2O$ over the catalyst was also collected under similar experimental condition but using $^{14}N_2$ rather than $^{15}N_2$

**Ex situ X-ray photoelectron spectroscopy (XPS) measurement and etching.** XPS measurement was performed on an ESCALAB 250Xi photoelectron spectrometer (Thermo Fisher Scientific) equipped with monochromatic Al-K$_\alpha$ source (K$_\alpha$ = 1,486.6 eV) and a charge neutralizer. The XPS binding energy was calibrated against the C1s peak at 284.6 eV of adventitious carbon. Prior to in situ measurements, XPS spectra of fresh sample were first acquired. Etching was carried out with the MAGCIS dual mode ion source, which can be operated as a monatomic argon ion source. The monatomic mode was operated at energy of 1000 eV.

In situ XPS measurement: The catalyst was treated at 350 °C for 2 h in a feed of $10\%H_2/Ar$ mixture (50 mL/min) in a pretreatment chamber attached to the spectrometer, followed by the acquisition of the Co2p, C1s, and N1s spectra. Finally, the sample was further in situ treated under a stream of $25\%N_2$–$75\%H_2$ at 350 °C for 2 h. Afterward, the Co2p, C1s, and N1s spectra were collected.

Computational method: First-principles calculations based on density functional theory (DFT) were performed using the Vienna ab initio simulation package (VASP) and the projected augmented wave (PAW) method[6]. The generalized gradient approximation Perdew–Burke–Ernzerhof (PBE) exchange-correlation functional was employed, and the DFT-D3 method of Grimme was employed to represent van der Waals interactions. The kinetic energy cutoff plane-wave expansion was set to 400 eV, and only Γ point was involved in the Brillouin zone integration. The convergence thresholds of the energy change and the maximum force for the structural optimizations were set to 10$^{-5}$ eV and 0.05 eV/Å, respectively. To investigate the catalytic reaction on Co–N–C more accurately, hybrid functional B3LYP-D3 was also employed to calculate the electronic energy based on the optimized structure of PBE-D3 functional[6]. The adsorption energies of $H_2$ on Co–N–C were calculated by $\Delta E = E(Co–N–C\_H_2) — E(Co–N–C) — E(H_2)$, Here $E(Co–N–C\_H_2)$, $E(Co–N–C)$, and $E(H_2)$ are the energies of optimized structures of Co–N–C-adsorbed $H_2$, Co–N–C, and $H_2$. Note that because $H_2$ adsorption weakened the interaction between phthalonitrile Co and carbon sphere, the $H_2$ adsorption energies on graphitic N and pyrrolic N are positive. However, there is the negative $H_2$ adsorption energies on pyridinic N. It is because the $H_2$ directly dissociates to two hydrogen atoms, one bonded to pyridinic N and the other coordinated with the Co atom, and the interaction between phthalonitrile Co and carbon sphere was not weakened.

## Data availability
The data that support the findings of this study are available from the corresponding author upon reasonable request.

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

## Acknowledgements

The work was supported by the National Science Fund for Distinguished Young Scholars of China (21825801), the National Natural Science Foundation of China (21972019), and Fujian Outstanding Youth Fund (2019J06011). We thank Prof. Bin Wang of Beijing Institute of Chemical Industry of Sinopec for providing the in situ equipment for EXAFS analysis, and Prof. Jingdong Lin of Xiamen University for providing the $^{15}N_2$ isotope gas.

## Author contributions

L.J. and X.W. proposed the research idea and supervised the entire project. X.P. performed the synthesis and catalytic measurements. X.P. and X.W. performed characterization. J.N. provides instrument Platform. W.C. and A.Z. carried out the model construction and density functional theory calculations. X.W. and G.L. performed in situ X-ray absorption fine-structure measurements. L.Z. performed the fit of EXAFS data. X.W. wrote the paper with significant contributions from C.A. All authors participated in the interpretation of results and made comments on the manuscript.

## Competing interests

The authors declare no competing interests.
