## [Peer Review File · Nature Communications]

Reviewers' comments:

Reviewer #1 (Remarks to the Author):

This is a very interesting and potentially important paper which examines an intriguing new catalytic system for ammonia synthesis. The Co/N/C catalyst shows high activity for this key reaction and the wealth of characterisation data reported clarifies that the activity is due to a single Co atom active site.

The paper reports DFT calculations which importantly suggest an associative mechanism, The calculations employ the now very standard and reliable PBC techniques available in the VASP code. Although the results are of considerable interest I have some concern over the use of the PBE functional which is known to be deficient in accurate modelling of catalytic processes. It would be helpful if at least some of the critical steps could also be examined using hybrid functionals. Also, we need much greater clarity on the actual model used for the catalyst. We need a detailed description of the surface structure employed and the location of the Co atoms. There are some figures in the SI but these also are insufficiently clear. Given attention to these points the paper can be accepted for publication.

Reviewer #2 (Remarks to the Author):

This is an interesting manuscript which details a new catalytic material and proposes an unusual mechanistic pathway in a very topical area. I recommend that the manuscript is accepted for publication following consideration of the following points:

- * error bars should be presented in relation to the catalytic data. The particle size histograms should be presented in terms of absolute counts
- * reference is made in the introduction to B5 sites in terms of the active site in the Fe system. Should it not be C7 sites in relation to bulk Fe? B5 sites are associated with the dispersed Ru system
- * upon inspection of the cycles presented in Figure 5, some consideration should be given to nomenclature since Mars-van Krevelen is associated with lattice species consumption and replenishment and this seems more like the second pathway which is termed looping. Looping is more generally associated with application of separated cycles operating under different conditions
- * can the alternative of a mechanism involving activation and spillover of H₂ in the Co site hydrogenating N₂ activated at the N vacancy be discounted?
- * the authors refer to, on a number of occasions, Co(NH₃)₆Cl₆ as a reference implying a compound with a Co oxidation state of +VI
- * in Figure 2(b) there is potential ambiguity. Is the mass of catalysts (g_{cat}), the total mass or just that of the Co component?
- * is the catalyst support resistant to methanation?
- * if the authors can strengthen the discussion of their ability to discount complications arising from overlapping mass fractions in their mass spec data, they should do so
- * better justification of the fitting parameters employed for the N1s XPS spectra is recommended
- * there are a number of issues related to language in the manuscript - for example it should be "operando" in the title, "Haber" in a number of places, "suite" on page 3, "Fourier" on page 5....
- * the apparently higher reactivity of ¹⁵N₂ versus ¹⁴N₂ as stated on page 11 is very difficult to rationalise, is it possible for the authors to provide an explanation for this observation?

Reviewer #3 (Remarks to the Author):

In this work the authors present a Co-based catalyst for the synthesis of NH₃ under soft conditions. They evaluate the catalytic performance of this material in terms of NH₃ synthesis rate, while the structural properties of the catalyst are investigated by X-ray based techniques, such as XRD, XPS, XAS, and transmission electron microscopy techniques. The study of the species involved in the reactions, and the interpretation of the NH₃ synthesis pathway, is completed by isotopic labelling methods and DFT calculations.

I consider this manuscript interesting, and the use of all these techniques shows a significant work, but honestly I am not completely convinced about the structural and mechanistic results.

The DFT section is not clear and well organized: it describes the synergistic effect of the Co-N_{3.5} site and a Co-N_x site embedded into a carbon matrix, without providing clear snapshots of the reaction steps. Figure S19 is incomprehensible, I don't see the H₂ dissociation and where the hydrogen atoms are bonded, while Figure 5a shows a pyrrolic nitrogen on which N₂ is hydrogenated, which I assume it is part of the Co-N_{3.5} site, not showed. The text does not help: in the sentence at page 14 "...H₂ directly dissociates to two hydrogen atoms, one bonded to pyridinic N and the other coordinated with the Co", so does Co directly coordinates H, or is hydrogen coordinated by a N atom in the CoN_{3.5} site? At the same page "it is deduced that the pyrrolic N serves to anchor a single Co atom", which pyrrolic N the authors mean? I assume the pyrrolic N of the CoN_{3.5} site, but it is not clear to me. During the reaction the coordination around the cobalt atom vary widely, and I wonder why the EXAFS spectra do not change. EXAFS should be able to see an increase or a change in the coordination number during NH₃ production. I am surprised that the CoPc square planar symmetry is maintained in operando conditions, which is confirmed by the presence of the pre-edge peak at 7116 eV in the XANES spectrum; the interactions with N₂ and/or the pyridinic nitrogen atoms should break this symmetry. In this regard, in the sentence at page 5 "The pre-edge peak of CoN₆/C at 7716 eV is assigned to the forbidden 1s→3d transition, corresponding to the Co-N structure", "Co-N" does not mean anything; the authors probably meant "CoPc square planar geometry with D_{4h} symmetry"

Other issues:

-This catalyst is considered by the authors as a single atom catalyst. It is not exactly like that: during the synthesis, the heating temperature of 500°C does not alter the CoPc structure (<https://doi.org/10.1016/j.fuel.2012.03.055>). This is clearly demonstrated by the presence of the pre-edge feature at 7116 eV in the XANES spectra, but also by the peaks in the Fourier transform of the EXAFS spectra (figure S16), that are characteristic of the CoPc (DOI:

10.1039/c6cp04022e) and still present in operando conditions. This is not a single atom catalyst, but simply cobalt phthalocyanine anchored on a nitrogen-containing carbon material. So, for example at page 8, the statement: "The excellent stability of the single-Co-atom sites in CoN₆/C can be attributed to the coordination between the isolated Co atoms and adjacent N atoms" loses its meaning, as well as the very concept of single-atoms catalyst along the entire manuscript.

-In Table S2 the errors on the EXAFS parameters are missing.

-The EXAFS fit of the ex-situ catalyst is not shown anywhere.

-I do not find the sample designation "CoN₆/C" correct. This notation makes us think that cobalt is coordinated by 6 nitrogen atoms.

-English needs to be carefully reviewed. There are typos and mistakes in the wording of some sentences.

Reviewers' comments:

Reviewer #1 (Remarks to the Author):

This is a very interesting and potentially important paper which examines an intriguing new catalytic system for ammonia synthesis. The Co/N/C catalyst shows high activity for this key reaction and the wealth of characterisation data reported clarifies that the activity is due to a single Co atom active site.

The paper reports DFT calculations which importantly suggest an associative mechanism, The calculations employ the now very standard and reliable PBC techniques available in the VASP code. Although the results are of considerable interest I have some concern over the use of the PBE functional which is known to be deficient in accurate modelling of catalytic processes. It would be helpful if at least some of the critical steps could also be examined using hybrid functionals. Also, we need much great clarity on the actual model used for the catalyst. We need a detailed description of the surface structure employed and the location of the Co atoms. There

are some figures in the SI but these also are insufficiently clear.

Given attention to these points the paper can be accepted for publication.

Response: Thank you for your positive comments.

As suggested, we employed the hybrid functional B3LYP-D3 to accurately describe the electronic energy based on optimized structures by PBE-D3 functional. The outcome is that the energy change of the critical step ($M_N_2_H_2 \rightarrow M_NHNH$) slightly decreases from 1.96 eV of PBE-D3 to 1.79 eV of B3LYP-D3. The related descriptions have been added to the section of computational method: “To investigate the catalytic reaction on $CoN_5/N-C$ more accurately, the hybrid functional B3LYP-D3 was also employed to calculate the electronic energy based on the optimized structure of PBE-D3 functional” (Page 23 of main text). Moreover, the free energy for the formation of NH_3 on $CoN_5/N-C$ has been updated in Figure 5a (Page 16).

Figure S24 Steps for the construction of $CoN_5/N-C$ model.

To represent the hollow N-doped porous carbon spheres, a surface with structure based on single-layer graphene with defect and dopant was constructed according to the description of Pan et.al. [*J. Am. Chem. Soc.* **2018**, 140, 4218–4221]. The Co atom is located at the center of phthalocyanine and coordinates with the N-dopant site on the carbon sphere. The structural information has been detailed in Supporting

Information as “First, we constructed a single layer graphene with 10*10*1 supercell, and then two connecting carbon atoms were defected to accommodate a nitrogen atom for the generation of pyrrolic N as site for coordination with the Co atom of phthalocyanine cobalt” (Page 26 of Supporting Information), and the results have been added in Figure S24”. Additionally, the snapshots of the reaction $N_2 + 3H_2 \rightarrow 2NH_3$ on $CoN_5/N-C$ based on DFT calculation has been added in Figure S25. The corresponding descriptions have been provided in the revised manuscript: “Moreover, the changes of free energy for the formation of NH_3 and the detailed reaction steps of $N_2 + 3H_2 \rightarrow 2NH_3$ on $CoN_5/N-C$ are illustrated in Figure 5a and Figure S25, respectively” (Page 15 of main text).

Figure S25 Snapshots of the reaction $N_2 + 3H_2 \rightarrow 2NH_3$ on $CoN_5/N-C$ based on DFT calculation (black: C, white: H; N: blue; Co: blue grey).

Reviewer #2 (Remarks to the Author):

This is an interesting manuscript which details a new catalytic material and proposes an unusual mechanistic pathway in a very topical area. I recommend that the manuscript is accepted for publication following consideration of the following points:

**(1) error bars should be presented in relation to the catalytic data. The particle size histograms should be presented in terms of absolute counts.*

Response: As suggested, error bars associated with NH_3 synthesis rate have been

provided, and the results are outlined below (Figure 2) and included in Figure 2a and Figure 2b of the revised manuscript. Meanwhile, the caption of Figure 2 has been added with “The data points and error bars represent the average and standard deviation based on multiple measurements on the same catalyst at different times over different batches of samples” (Page 7 of main text).

Figure 2 NH₃ synthesis rate at different temperatures for CoN₅/N-C, Co/C and N-C under 1 MPa and (b) NH₃ synthesis rate of select catalysts for NH₃ synthesis at 350 °C under 1 MPa (The data points and error bars represent the average and standard deviation based on multiple measurements on the same catalyst at different times over different batches of samples).

Moreover, we have provided the statistical results of Co particle size based on AC-STEM results in terms of ~250 particles, and the results have been outlined below and included in Figure S2 (Page 5 of Supporting Information). Thanks.

Figure 3 (a–c) Three representative aberration-corrected high-angle annular dark field-scanning transmission electron microscopy images of fresh CoN₅/N-C catalyst, all of them showing atomically dispersed Co atoms, and (d) histogram of Co particle size distribution over CoN₅/N-C.

(2) *reference is made in the introduction to B₅ sites in terms of the active site in the Fe system. Should it not be C₇ sites in relation to bulk Fe? B₅ sites are associated with the dispersed Ru system*

Response: We have removed the description of B₅ sites associated with bulk Fe in the introduction section, and the sentence “through theoretical studies Li et al.⁶ demonstrated that if metal Fe is in the form of individual Fe₃ clusters rather than in groups of B₅ sites” has been revised to “through theoretical studies Li et al.⁶ demonstrated that if metal Fe is in the form of individual Fe₃ clusters rather than in groups of C₇ sites” (Page 2 of main text). Thanks.

(3) *upon inspection of the cycles presented in Figure 5, some consideration should be given to nomenclature since Mars-van Krevelen is associated with lattice species consumption and replenishment and this seems more like the second pathway which is termed looping. Looping is more generally associated with application of separated cycles operating under different conditions*

Response: Thank you for reminding. The caption “Eley-Rideal/Mars-Van Krevelen

(E-R) mechanism” of Figure 5b has been revised to “Eley-Rideal (E-R) mechanism” (Page 16). Thanks.

**(4) can the alternative of a mechanism involving activation and spillover of H₂ in the Co site hydrogenating N₂ activated at the N vacancy be discounted?*

Response: The results of our experimental and theoretical studies support that the N₂ activated at the N vacancy is hydrogenated by the H₂ activated at the Co site. First, the signal of NH₃ formation synchronizes with the pulse of H₂ introduction, suggesting that the hydrogenation of the N species on CoN₅/N-C is facile. In addition, the formation of NH₃ leaves behind an anionic N vacancy. Second, the isothermal ¹⁵N₂ isotopic labeling experiments confirm that the nitrogen consumed can be replenished by gaseous nitrogen, indicating that gaseous nitrogen is activated at the anionic N vacancy. Then with the feeding of 10%H₂/Ar, the NH₃ signal can be largely restored to its previous value, evidencing that the N₂ activated at the N vacancy is hydrogenated by the H₂ activated at the Co site. Moreover, the DFT calculation shows that only surface pyridinic N species could be readily activated by adsorbed H₂ to produce NH₃, while gaseous N₂ can be activated by anionic N vacancies. To be noted, the amount of NH₃ produced in this way (1.12 mmol_{NH₃} per gram of catalyst, as shown in Figure S15) is only 25.8% of the total amount of NH₃ generated (4.34 mmol_{NH₃} per gram of catalyst, as shown in Figure 2a).

**(5) the authors refer to, on a number of occasions, Co(NH₃)₆Cl₃ as a reference implying a compounds with a Co oxidation state of +VI*

Response: Sorry for the careless mistake. The caption of “Co(NH₃)₆Cl₆” reference sample of Figure 1i & Figure 1j has been revised to “Co(NH₃)₆Cl₃”. The purpose of using Co(NH₃)₆Cl₃ as reference sample was to confirm the existence of Co-N bond in the case of CoN₅/N-C. The position of the absorption edge could be used as an indicator to estimate the valence states of Co species. According to the magnified XANES results (inset image of Figure 1i), the absorption edge position of CoN₅/N-C is located between that of CoO and Co(NH₃)₆Cl₃, suggesting that the single Co atom

carries positive charge with valence state between +2 and +3. Meanwhile, the XPS spectra of $\text{Co}2p_{3/2}$ also confirm that the oxidation state of Co is between +2 and +3.

The related descriptions have been provided in the text as “The position of the absorption edge could be used as an indicator to estimate the valence states of Co species. According to the magnified XANES results (Figure 1i), the absorption edge position of $\text{CoN}_5/\text{N-C}$ is located between that of CoO and $\text{Co}(\text{NH}_3)_6\text{Cl}_3$, suggesting that the single Co atom carries positive charge with valence state between +2 and +3” (Pages 5–6). Thanks.

**(6) in Figure 2(b) there is potential ambiguity. Is the mass of catalysts (gcat), the total mass or just that of the Co component?*

Response: In Figure 2b, NH_3 synthesis rate was calculated in term of total catalyst mass, while the NH_3 synthesis rate illustrated in Table S3 was calculated in the form of per-Co-active-site. Thanks.

**(7) is the catalyst support resistant to methanation?*

Response: Good question. Our previous studies have shown that carbon methanation readily occurs over Ru/C catalysts but not over Co-based catalysts. In the present studies, we found that the carbon support does not undergo methanation when the temperature and pressure are kept at 400 °C and 1 MPa, respectively. In order to evaluate whether the $\text{CoN}_5/\text{N-C}$ catalyst would undergo methanation during NH_3 synthesis, the outlet CH_4 concentration was monitored by an on-line GC-Mass equipment (GCMS-QP2010 SE), and the results have been added as Figure S11 in the Supporting Information.

Figure S11 The outlet CH₄ concentration as a function of time during NH₃ synthesis over CoN₅/N-C sample at different temperatures (conditions: WHSV= 60 000 ml·g⁻¹·h⁻¹, 1 MPa).

The relevant descriptions have been provided in the main text with “During NH₃ synthesis at either 350 or 400 °C for 100 h under 1MPa, the outlet CH₄ concentration is negligibly low (Figure S11), suggesting that under the adopted conditions the carbon support is highly stable” (Page 8 of main text). As for the description of the methanation experiments, it has been added in the Method section: “In the determination of the possibility of methanation, 0.25 g of CoN₅/N-C was exposed to a flow of 10%H₂/Ar at 400 °C for 4 h. In another case, the conditions were under a 25%N₂-75%H₂ mixture at WHSV of 60 000 ml·g⁻¹·h⁻¹ and pressure of 1MPa. In both experiments, the outlet CH₄ concentrations were measured using an on-line GC-Mass (GCMS-QP2010 SE).” (Page 20 of main text). Thanks.

**(8) if the authors can strengthen the discussion of their ability to discount complications arising from overlapping mass fractions in their mass spec data, they should do so.*

Response: Kind reminding. According to the results of ¹⁵N₂ isotopic labeling experiment over CoN₅/N-C at 350 °C in the presence of ¹⁵N₂ and H₂, the intensity of the m/z=16 signal is almost constant while the m/z=17 signal changes with the change of ¹⁵N₂ concentration (Figure S21). Therefore, the m/z=16 and m/z=17 signal can be assigned to ¹⁴NH₂ and ¹⁵NH₂, respectively. The corresponding descriptions have been

added in the revised text with “The $m/z=16$ signal could be due to $^{14}\text{NH}_2$ or ^{15}NH , and because its intensity is almost constant when there is a decrease of $^{15}\text{N}_2$, it is reasonable to assign it to $^{14}\text{NH}_2$. As for the $m/z=17$ signal, its intensity changes with the change of $^{15}\text{N}_2$ concentration (Figure S21); it is hence reasonable to assign it to $^{15}\text{NH}_2$ ” (Page 13 of main text). Thanks.

** (9) better justification of the fitting parameters employed for the N1s XPS spectra is recommended*

Response: Thanks for your good suggestion. The N1s XPS spectra of $\text{CoN}_5/\text{N-C}$ have been re-fitted and the results displayed in Figure 4d and Table S4. Meanwhile, the related N1s description has been revised: “four N species can be identified (detailed parameters provided in Table S4). They are graphitic (400.7 eV), pyrrolic (399.4 eV), pyridinic (398.9 eV) and N-oxide (404.3 eV)²². The surface composition of pyridinic N species is 38.9%. The N1s spectrum recorded after exposure to 10% H_2/Ar at 350 °C for 2 h shows a surface pyridinic N composition of 24.4%. In the case of exposing the catalyst to 25% N_2 -75% H_2 at 350 °C for 2 h, the surface composition of pyridinic N species returns back to 38.7%” (Page 14).

** (10) there are a number of issues related to language in the manuscript - for example it should be "operando" in the title, "Haber" in a number of places, "suite" on page 3, "Fourier" on page 5....*

Response: Sorry for the typos and we have made the necessary corrections. Meanwhile, we have carefully checked the language of the manuscript. Thanks.

**(11) the apparently higher reactivity of $^{15}\text{N}_2$ versus $^{14}\text{N}_2$ as stated on page 11 is very difficult to rationalise, is it possible for the authors to provide an explanation for this observation?*

Response: Good question. We believe that the unusual $^{15}\text{NH}_3$ intensity may be originated from the interference of H_2O in the TCD system during pipeline switching. Specifically, H_2 -Ar mixture and Ar gases passed through the preparation pipeline

linked with the 2920 instrument, while $^{14}\text{N}_2$ or $^{15}\text{N}_2$ passed through the carrier pipeline connected with the 2920 instrument. That is to say, during the experimental processes, the pipeline should be switched from that of preparation gas to the carrier one when the gas is switched from Ar to $^{14}\text{N}_2$ or $^{15}\text{N}_2$. Before the isothermal surface labeling experiment, we used $^{14}\text{N}_2$ gas to remove the air in the carrier gas pipeline and this process lasted for about 15 min. However, this purging step was not performed in the case of $^{15}\text{N}_2$, mainly owing to the high price of $^{15}\text{N}_2$. Therefore, the presence of air in the carrier gas pipeline would result in the reaction of oxygen species with H_2 adsorbed on the catalyst surface to form H_2O , giving rise to the H_2O signal ($m/z=18$) and causing interference to the $^{15}\text{NH}_3$ signal intensity.

In order to verify this point, we added an additional procedure before the isothermal $^{15}\text{N}_2$ isotopic labeling experiment. That is to use $^{15}\text{N}_2$ (30 mL/min) to purge the carrier gas pipeline for 15 min. In such a case, the signal intensity of $^{15}\text{NH}_3$ is similar to that of $^{14}\text{NH}_3$ in the isothermal $^{15}\text{N}_2$ isotopic labeling experiment. On the basis of the above phenomenon, the results displayed in Figure 3c have been updated with the signal intensity of $^{15}\text{NH}_3$ similar to that of $^{14}\text{NH}_3$. Accordingly, the related description in the original text: “It is noted that despite similar in experimental procedure and conditions, the $^{15}\text{NH}_3$ signal of Figure 3c is much higher than the $^{14}\text{NH}_3$ signal of Figure 3b in intensity. The results suggest that in the adopted experimental setting, $^{15}\text{N}_2$ shows higher reactivity than $^{14}\text{N}_2$ for NH_3 production.” has been deleted.

Meanwhile, the description for the isothermal ^{15}N isotopic labeling experiment has been updated in the Method section: “Before the experiment, the carrier gas pipeline was purged with $^{15}\text{N}_2$ (30 mL/min) for 15 min.” (Page 20). Additionally, raw data of various mass signals during total $^{15}\text{N}_2$ isotopic labeling processes have been provided in the data package. Thanks.

Reviewer #3 (Remarks to the Author):

In this work the authors present a Co-based catalyst for the synthesis of NH_3 under soft conditions. They evaluate the catalytic performance of this material in terms of

NH₃ synthesis rate, while the structural properties of the catalyst are investigated by X-ray based techniques, such as XRD, XPS, XAS, and transmission electron microscopy techniques. The study of the species involved in the reactions, and the interpretation of the NH₃ synthesis pathway, is completed by isotopic labelling methods and DFT calculations.

I consider this manuscript interesting, and the use of all these techniques shows a significant work, but honestly I am not completely convinced about the structural and mechanistic results.

Response: Thank you for your positive comments. In response to the comments, we have performed additional experiments.

(1) *The DFT section is not clear and well organized: it describes the synergistic effect of the Co-N_{3.5} site and a Co-N_x site embedded into a carbon matrix, without providing clear snapshots of the reaction steps. Figure S19 is incomprehensible, I don't see the H₂ dissociation and where the hydrogen atoms are bonded, while Figure 5a shows a pyrrolic nitrogen on which N₂ is hydrogenated, which I assume it is part of the Co-N_{3.5} site, not showed. The text does not help: in the sentence at page 14 "...H₂ directly dissociates to two hydrogen atoms, one bonded to pyrindinic N and the other coordinated with the Co", so does Co directly coordinates H, or is hydrogen coordinated by a N atom in the CoN_{3.5} site? At the same page "it is deduced that the pyrrolic N serves to anchor a single Co atom", which pyrrolic N the authors mean? I assume the pyrrolic N of the CoN_{3.5} site, but it is not clear to me.*

Response: (a) We have updated Figure 5a with more detailed reaction pathways to clarify the catalytic reaction steps of ammonia synthesis on CoN₅/N-C catalyst.

Figure S23 H₂ dissociation on CoN₅/N-C (black: C, white: H; N: blue; Co: blue grey).

To better describe the H₂ dissociation on CoN₅/N-C, we have replaced Figure S19 with Figure S23. The H₂ molecule spontaneously dissociates into two hydrogen atoms, one bonded to pyrrolic N and the other coordinated with the Co; and the overall reaction is exothermic (-2.87 eV). Moreover, the detailed reaction steps of N₂ + 3H₂ → 2NH₃ reaction on CoN₅/N-C have been added as shown in Figure S25.

Figure S25 Snapshots of the reaction $\text{N}_2 + 3\text{H}_2 \rightarrow 2\text{NH}_3$ on CoN₅/N-C based on DFT calculation.

Meanwhile, the related descriptions have been added in the main text: “The steps to construct the model of CoN₅/N-C for NH₃ synthesis is shown in Figure S24 (Please see Supporting Information for more details), and it can be observed that the Co atom is located at the center of phthalocyanine and coordinates with the N-dopant site on carbon sphere. The changes of free energy for the formation of NH₃ and the detailed reaction steps of N₂ + 3H₂ → 2NH₃ reaction on CoN₅/N-C are illustrated in Figure 5a and Figure S25, respectively” (Page 15). Thanks.

(2) During the reaction the coordination around the cobalt atom vary widely, and I wonder why the EXAFS spectra do not change. EXAFS should be able to see an increase or a change in the coordination number during NH₃ production. I am surprised that the CoPc square planar symmetry is maintained in operando conditions, which is confirmed by the presence of the pre-edge peak at 7116 eV in the XANES spectrum; the interactions with N₂ and/or the pyridinic nitrogen atoms should break this symmetry. In this regard, in the sentence at page 5 “The pre-edge peak of

CoN₆/C at 7716 eV is assigned to the forbidden 1s→3d transition, corresponding to the Co-N structure”, “Co-N” does not mean anything; the authors probably meant “CoPc square planar geometry with D_{4h} symmetry”.

Response:

For the question “During the reaction the coordination around the cobalt atom vary widely, and I wonder why the EXAFS spectra do not change. EXAFS should be able to see an increase or a change in the coordination number during NH₃ production”, we give the following response:

(a) *In situ* XAS spectra of the catalysts in a 10%H₂/He or N₂-H₂ (V_{N₂}:V_{H₂}=1:3) atmosphere were previously fitted by a single layer method, and the results only give the first Co-N coordination number. In the revised manuscript, we fitted the *in-situ* EXAFS spectra with two layers, and the results are provided in Figure S18–S19 (below) and Table S2. The coordination number of first Co-N shell varies in the range of 3.2–3.5 under 10%H₂/He atmosphere with different exposure times. To be noted, because the fitting error by itself was about 20%, the change of coordination number in the case of first layer Co-N does not explain much. Nonetheless, that of the second Co-N shell in CoN₅/N-C decreases from 1.5 over fresh catalyst to 0.9 after exposure to 10%H₂/He mixture atmosphere for 60 min (Figure S18 and Table S2), suggesting the N species could be easily activated by gas-phase H₂.

Figure S18 (A-D) *In-situ* EXAFS results of CoN₅/N-C catalyst: two layers fitting Co K-edge EXAFS curve of CoN₅/N-C exposed at 350 °C in the presence of 10%H₂/He with different exposure times.

Table S2 EXAFS fitting parameters at the Co K-edge for CoN₅/N-C.

Sample	Shell	N ^a	R (Å) ^b	σ^2 (Å ² ·10 ³) ^c	ΔE_0 (eV) ^d	R factor (%)
Fresh	Co-N	3.5	1.90	2.6	6.0	0.7
	Co-N	1.5	2.35			
10%H ₂ /He for 15min	Co-N	3.4	1.90	2.3	5.8	0.2
	Co-N	1.0	2.35			
10%H ₂ /He for 30 min	Co-N	3.2	1.91	1.7	6.0	0.2
	Co-N	0.9	2.35			
10%H ₂ /He for 60 min	Co-N	3.5	1.90	3.0	5.1	0.7
	Co-N	0.9	2.35			
N ₂ -H ₂ mixture for 60 min	Co-N	3.5	1.91	2.4	6.0	0.7
	Co-N	1.3	2.35			

^aCN: coordination numbers;

^bR: bond distance;

^c σ^2 : Debye-Waller factnd distors;

^d ΔE_0 : the inner potential correction.

R factor: goodness of fit.

The errors range of N and σ^2 are 20%, and the precision range of R is ± 0.03 Å.

(b) Despite the pyridine N could be consumed by H₂ (resulting in the decrease of

coordination number), the N species can be replenished by gaseous N₂ as confirmed by the *in situ* XPS and ¹⁵N₂ isotopic labeling experiments. Therefore, the coordination numbers of both Co-N shells do not fluctuate significantly under N₂-H₂ mixture atmosphere. That is to say, through two-layer fitting, one can see that the coordination number of the second Co-N shell decreases slightly in a H₂ atmosphere. However, in a N₂ + H₂ (V_{N2}:V_{H2}=1:3) atmosphere, the N species is consumed and then replenished, and the coordination number remains stable. The related descriptions have been added in the revised manuscript (Page 10 of main text). Thanks.

Figure S19 Co K-edge EXAFS fitting curve of CoN₅/N-C exposed at 350 °C in the presence of N₂-H₂ mixture (V_{N2}:V_{H2}=1:3).

For the questions “*In this regard, in the sentence at page 5 “The pre-edge peak of CoN₆/C at 7716 eV is assigned to the for bidden 1s→3d transition, corresponding to the Co-N structure”, “Co-N” does not mean anything; the authors probably meant “CoPc square planar geometry with D_{4h} smmetry”, we give the following response:*

Thank you for the good suggestion. Cobalt phthalocyanine has a D_{4h} symmetry with XANES exhibiting two pre-edge transitions at 7709 eV and 7716 eV. The

pre-edge peak at 7716 eV is associated with a fingerprint of CoN₄ planar structure (*J. Phys. Chem.* **1992**, *96*, 10898–10905). The sentence has been corrected as “The pre-edge peak of CoN₅/N-C at 7716 eV is assigned to the forbidden 1s→3d transition, which is considered as a fingerprint of Co-N_x planar structure” (Page 5). The revised paper has been updated with the related references (i.e., Refs. 10, 18 and 19). Thanks.

Other issues:

(3) *This catalyst is considered by the authors as a single atom catalyst. It is not exactly like that: during the synthesis, the heating temperature of 500°C does not alter the CoPc structure (<https://doi.org/10.1016/j.fuel.2012.03.055>). This is clearly demonstrated by the presence of the pre-edge feature at 7116 eV in the XANES spectra, but also by the peaks in the Fourier transform of the EXAFS spectra (figure S16), that are characteristic of the CoPc (DOI: 10.1039/c6cp04022e) and still present in operando conditions. This is not a single atom catalyst, but simply cobalt phthalocyanine anchored on a nitrogen-containing carbon material. So, for example at page 8, the statement: “The excellent stability of the single-Co-atom sites in CoN₆/C can be attributed to the coordination between the isolated Co atoms and adjacent N atoms” loses its meaning, as well as the very concept of single-atoms catalyst along the entire manuscript.*

Response: On the basis of four factors we believe the reported catalyst is with single-Co-atom sites:

First, it is true that it is a case of “CoPc anchored on an N-doped carbon support” but it is also a case of cobalt highly dispersed on the support in the form of single atoms. When the Co particle size is reduced to that of single Co atoms for homogeneous distribution of active sites, it reaches the ultimate small-size limit for metal particles, viz. that of single-atom catalyst (SAC) (*Acc. Chem. Res.* **2013**, *46*, 1740–1748; *Acc. Chem. Res.* **2019**, *52*, 656–664). In such a case, the catalytically active metal is exclusively dispersed as single atoms on the support. Meanwhile, these individual Co atoms should be stabilized by covalent coordination or ionic

interactions with neighbouring surface atoms (*Nat. Catal.*, **2018**, 1, 385–397; *Nature Reviews Chemistry*, **2018**, 65–81). Currently, the single atom nature is determined by the absence of Co-Co bonds, as usually evidenced with AC-STEM and EXAFS (*Nat. Commun.* **2017**, 8, 957; *J. Am. Chem. Soc.* **2017**, 139, 17269–17272; *ACS Energy Lett.* **2019**, 4, 1816–1822; *Angew. Chem. Int. Ed.* **2016**, 55, 10800–10805; *Nat. Commun.*, **2018**, 9, 3861; *Nat. Commun.*, **2018**, 9, 3197). These single-atom sites were stabilized with neighbouring atoms, such as C, N or O to form Co-N, Co-C or Co-O coordination (*Angew. Chem. Int. Ed.* **2018**, 57, 11262–11266; *Adv. Mater.* **2018**, 30, 1706758; *Chem. Sci.*, **2016**, 7, 5758–5764). In the present study, the rigid planar macrocycle structure of CoPc molecules and the steric hindrance effect of N-doped carbon hollow nanospheres are responsible for the dispersion of Co-N_x species at an atomic level (*Adv. Sci.* **2019**, 6, 1801103).

Second, in the present study there are evidences to support that cobalt exists in the form of single atoms. In the HR-TEM images (Figure 1b), despite much efforts we could not detect any sight of Co particles or clusters across a large sample area. The very fact that the cobalt cannot be detected by HR-TEM technique implies that the cobalt species must be highly dispersed as either tiny clusters or single atoms. In the measurements using aberration-corrected high-angle annular dark-field scanning transmission electron microscopy (AC-STEM) of sub-angstrom resolution, uniformly dispersed atomic Co sites (white dots, Figure 1g & Figure 1h) are clearly distinguished as bright spots throughout the area under investigation. Because the same can be observed in different sample areas, it can be deduced that there is uniform dispersion of atomic Co throughout the N-doped carbon hollow spheres (Figure S2a & Figure S2b). Meanwhile, the statistical results of Co size in terms of ~250 particles (Figure S2c) show that the particle size of these Co species is ~1.3 Å (Figure S2d), confirming that Co predominantly exists as single atoms rather than in the form of tiny clusters. Finally, *ex-situ* EXAFS and *in situ* EXAFS spectra show no Co-Co coordination peak in comparison with Co foil reference (Figure S1j), indicating the absence of Co metallic clusters in the catalyst. Based on the above characterization results, it is reasonable to conclude that CoN₅/N-C is a single-atom

catalyst.

Third, similar studies have been reported. When CoPc or FePc precursors were inserted in N-C or graphene supports at relatively low temperature such as below 800 °C, Co or Fe uniformly dispersed in the form of single atoms (*J. Am. Chem. Soc.* **2018**, 140, 4218–4221, *Sci. Adv.* **2019**, DOI: 10.1126/sciadv.aaw2322; *J. Power Sources.* **2011**, 196, 2519–2529. *Adv. Sci.* **2019**, 6, 1801103; *ACS Catal.* **2019**, 9, 6252–6261). Furthermore, the results of XANES analysis revealed that SAC-Co contained near-edge structures similar to those of the original CoPc or FePc but totally different from those of Co or Fe foil. Based on the fact that there is no Co-Co or Fe-Fe coordination, it was considered that Co or Fe existed in the form of single atoms (*J. Am. Chem. Soc.* **2018**, 140, 4218–4221) or SAC-Fe (*Sci. Adv.* **2019**, DOI: 10.1126/sciadv.aaw2322; *J. Am. Chem. Soc.* **2017**, 139, 10790–10798. *Nat. Commun.* **2019**, 10, 1278).

Fourth, our studies show that there is interaction between CoPc and N-doped carbon support. The quantitative coordination configuration of Co atoms can be obtained by EXAFS fitting, and the Co-N coordination number (CN) of CoN₅/N-C is 5.0 (3.5+1.5), while that of CoPc is 4, indicating the presence of coordination between single Co atom in CoPc and N atom in N-doped carbon hollow spheres. It is noted that CoPc shows a Co-N₄ square planar geometry with D_{4h} symmetry, and the valence state of Co species is Co²⁺ (*ACS Catal.* **2019**, 9, 2521–2531), whereas in our study the XPS Co 2p spectrum (Figure S5) of CoN₅/N-C shows a hetero valence state of Co²⁺ and Co³⁺. These results proved that the Co state and coordination environment of CoPc and that of CoN₅/N-C are not the same.

Based on the above discussion, it is reasonable to believe that the CoN₅/N-C catalyst is comprised of atomically dispersed cobalt. Meanwhile, more relevant descriptions of AC-STEM and EXAFS spectra associated with single Co atoms have been added in the text. Thanks.

(4) *In Table S2 the errors on the EXAFS parameters are missing.*

The EXAFS fit of the ex-situ catalyst is not shown anywhere.

Response: Kind reminding. The error range on the EXAFS parameters have been added in Table S2 with “The error range of N and σ^2 are 20%, and the precision range of R is $\pm 0.03 \text{ \AA}$ ” (Page 29 of Supporting Information).

The EXAFS fit of the ex-situ catalyst has been added and the result are provided in Figure S6. The corresponding description has been added in the main text (Page 6).

Thanks.

Figure S6 Co K-edge EXAFS fitting curve of fresh CoN₅/N-C catalyst.

(5) *I do not find the sample designation “CoN₆/C” correct. This notation makes us think that cobalt is coordinated by 6 nitrogen atoms.*

Response: We thank the reviewer for pointing that out. The loading of Co is 3.73 wt% and the N/Co molar ratio is 6.04. Nonetheless, according to the results of two-layer fitting of EXAFS, the cobalt atom is coordinated with five nitrogen atoms. Thus the designation of sample “CoN₆/C” has been revised to CoN₅/N-C in the main text and Supporting Information.

(6) *English needs to be carefully reviewed. There are typos and mistakes in the*

wording of some sentences.

Response: We have carefully checked the English writing of the entire paper and made the necessary corrections. Thanks.

Reviewers' comments:

Reviewer #1 (Remarks to the Author):

The authors have responded carefully and thoroughly to my previous recommendations for modification of the paper. As commented previously it is a good and original study in an important area of catalytic science and I am happy to recommend the paper for publication

Reviewer #2 (Remarks to the Author):

The authors have addressed the various points raised in a generally satisfactory manner. It is my view that the study will stimulate further work in this highly topical area. Accordingly, I recommend publication of the manuscript but I do suggest that the following are addressed:

- page 2 - to avoid ambiguity, I suggest referring to Fe₃O₄ and Fe_{1-x}O "derived" catalysts
- Figure S2(d) ordinate axis should be labelled "Particle number"
- page 7 missing word, textural "properties"
- page 8, check the quoted turnover frequencies versus Figure S10 where there is a factor of 10⁻³ quoted with respect to the turnover frequency axis
- page 19, spelling should be "Barrett"

Reviewer #3 (Remarks to the Author):

The EXAFS analysis is not self-evident: the technique makes it impossible to elucidate the coordination with such accuracy, simply because of the high error in the coordination numbers. It makes no sense to rely on a coordination number of 1.5 versus 0.9 when the error is 20%. I asked to specify the errors on the EXAFS parameters, but I just see estimation on the coordination numbers and R, while there are no errors on the Debye-Waller factors and E₀. In particular, errors have to be provided for every single parameter, it is not possible to have the same error on the entire set (for example $\pm 0.03\text{\AA}$ for all the distances). It is not possible to have the same Debye-Waller for the two Co-N bond distances, and further clarification: 1) Debye Waller units are in this case 10⁻³Å²; 2) in the legend of table S2 what does "factnd distors" mean? 3) "fit with two layers" layers is not the right word, "shells" has to be used. I have the impression that the authors do not handle the XAS technique at an appropriate level, and if you are not careful in the fitting procedure, you can determine different sets of parameters (then different results). Anyway, it is not possible to consider this analysis the background of the discussion and claim that CoN₅ is the right formula. Again, as far as the nature of these active sites is concerned, I thank the authors for all the references on the subject, but I did not mean that I suspect the presence of Co nanoparticles. I agree that the sites are dispersed, but this is not an active site in the form of CoN_x moieties, because the structure of CoPc is maintained. This is a clear spectroscopic evidence, and I wonder why the authors used the XAS spectra of CoO and hexaammincobalt(III) chloride as references and not the Co phthalocyanine spectrum, which is probably the most similar system in terms of structure, because the position of the absorption edge also depends on the specific compound. Moreover, by comparing the spectra of the catalyst and the CoPc I am pretty sure to see a very similar spectrum. If XPS finds a valence state of 2+ and 3+ for cobalt, that is not saying much, because XPS is a surface technique, and at the surface you may have oxidized species.

Reviewers' comments:

Reviewer #1 (Remarks to the Author):

The authors have responded carefully and thoroughly to my previous recommendations for modification of the paper. As commented previously it is a good and original study in an important area of catalytic science and I am happy to recommend the paper for publication

Response: Thank you for your positive comments.

Reviewer #2 (Remarks to the Author):

The authors have addressed the various points raised in a generally satisfactory manner. It is my view that the study will stimulate further work in this highly topical area. Accordingly, I recommend publication of the manuscript but I do suggest that

the following are addressed:

Response: Thank you for your positive comments.

- page 2 - to avoid ambiguity, I suggest referring to Fe_3O_4 and $Fe_{1-x}O$ "derived" catalysts

Response: As suggested, we have made changes accordingly.

- Figure S2(d) ordinate axis should be labelled "Particle number"

Response: Ordinate axis in Figure S2d has been revised to "Particle number", see page 5. Thanks.

- page 7 missing word, textural "properties"

Response: The word "properties" has been added to the words which is now becomes "the corresponding structure and textural properties". See page 8, thanks.

- page 8, check the quoted turnover frequencies versus Figure S10 where there is a factor of 10^{-3} quoted with respect to the turnover frequency axis.

Response: We have made corrections accordingly, see page 8. Thanks.

- page 19, spelling should be "Barrett"

Response: "Barett-Joyner-Halenda" has been revised to "Barrett-Joyner-Halenda", see page 19. Thanks.

Reviewer #3 (Remarks to the Author):

The EXAFS analysis is not self-evident: the technique makes it impossible to elucidate the coordination with such accuracy, simply because of the high error in the coordination numbers. It makes no sense to rely on a coordination number of 1.5 versus 0.9 when the error is 20%. I asked to specify the errors on the EXAFS parameters, but I just see estimation on the coordination numbers and R, while there

are no errors on the Debye-Waller factors and E^0 . In particular, errors have to be provided for every single parameter, it is not possible to have the same error on the entire set (for example $\pm 0.03\text{\AA}$ for all the distances). It is not possible to have the same Debye-Waller for the two Co-N bond distances, and further clarification.

Response: We thank the reviewer for the insightful comments. As suggested, errors for every single parameter have been provided in Table S2. The Debye-Waller factor given in previous Table S2 was associated with the first Co-N shell, and now Debye-Waller factors for the first and second Co-N shells have been provided in the annotation of Table S2.

Additionally, as pointed out by the reviewer, because the error of coordination number in the fitting process is 20%, the decrement of Co-N coordination number from 1.5 to 0.9 cannot be taken to make any reliable deduction. For this reason, we have deleted the statement that N species was consumed on the basis of the change of Co-N coordination number in the presence of 10% H_2/He atmosphere. It is worth noting that the consumption of N species is solid in isothermal surface reaction as well as in a series of H_2 pulse and a suite of $^{15}\text{N}_2$ isotopic labeling and exchange experiments. In the revised manuscript, the conclusions based on the EXAFS measurements are the absence of Co-Co coordination and Co coordination with N species during NH_3 synthesis reaction.

Debye Waller units are in this case 10^{-3}\AA .

Response: We thank the reviewer for pointing this out. Debye Waller units have been changed to 10^{-3}\AA in the annotation of Table S2.

in the legend of table S2 what does “factnd distors” mean?

Response: Sorry for the typos, the legend of Table S2 has been revised to “Debye-Waller factors”. Thanks.

“fit with two layers” layers is not the right word, “shells” has to be used. I have the impression that the authors do not handle the XAS technique at an appropriate level,

and if you are not careful in the fitting procedure, you can determine different sets of parameters (then different results). Anyway, it is not possible to consider this analysis the background of the discussion and claim that CoN₅ is the right formula.

Response: Thanks the reviewer for reminding. The phrase of “*fit with two layers*” has been revised to “fit with two shells”. As to “*different sets of parameters, and then obtain different results*”, we ensured that the parameter settings of different samples are the same in the fitting process. Meanwhile, the first and second coordination numbers of Co-N are 3.5 ± 0.7 and 1.5 ± 0.3 , respectively. Since there was a 20% error in the fitting process, we agree with the reviewer that CoN₅ may not be the right format. To avoid disputes on the composition of catalysts, the designation of sample “CoN₅/N-C” has been revised to Co-N-C throughout the main text and Supporting Information.

Again, as far as the nature of these active sites is concerned, I thank the authors for all the references on the subject, but I did not mean that I suspect the presence of Co nanoparticles. I agree that the sites are dispersed, but this is not an active site in the form of CoN_x moieties, because the structure of CoPc is maintained. This is a clear spectroscopic evidence.

Response: Atomically dispersed Co was considered as active sites during NH₃ synthesis. It is worth noting that these single Co atoms should be stabilized by covalent coordination or ionic interactions with neighboring atoms. In the present study, the single atom of Co was stabilized by neighboring N atoms in the form of Co-N_x. In other words, a single-atom cobalt active site cannot stably exist by itself, it should be stabilized by N atoms. As a result, we previously described the active site as Co-N_x. It could be more rigorous to present the description as “single Co active sites in the form of Co-N_x”. Accordingly, we have adopted such a description in the revised manuscript. Thanks.

I wonder why the authors used the XAS spectra of CoO and hexaamminecobalt(III) chloride as references and not the Co phthalocyanine spectrum, which is probably the

most similar system in terms of structure, because the position of the absorption edge also depends on the specific compound. Moreover, by comparing the spectra of the catalyst and the CoPc I am pretty sure to see a very similar spectrum.

Response: Thanks for your suggestion. XANES and EXAFS measurements of cobalt phthalocyanine (CoPc) reference have been supplemented and the results are provided in Figure 1i and Figure 1j, respectively. The corresponding descriptions have been updated in the main text (page 5). Meanwhile, the related references have been added to the main text (Ref. 20 and 21, see page 26).

In order to make a clear comparison between the structures of the Co-N-C catalyst and the CoPc reference, XANES spectra of both of them have been added in Figure S5. It is noted that they are similar in structure (Figure S5), consistent with the reviewer's anticipation. Nonetheless, a closer inspection reveals that there are obvious differences in features (a, b, c, d, see Figure S5).

Figure S5 Normalized Co K-edge XANES spectra of Co-N-C and CoPc reference.

Table 1 Energies and assignments for features observed in the Co-K XANES spectrum of CoPc.

Features	Energy/eV	Assignment
a	7709.5	1s → 3d (p-d hybridization)
b	7716.5	1s → 4p _z + ligand hole
c	7729.8	1s → 4p _{xy} + ligand hole
d	7736.9	1s → 4p _{xy}

Specifically, the Co-K edge XANES spectrum of CoPc (Figure S5) exhibits several transitions (labeled as a, b, c and d); their energy values are listed in Table 1. To be noted, the transition b which is observed for compounds with square-planar coordination is a fingerprint of the Co-N₄ structure and any modification of the environment would greatly affect this transition (*J. Phys. Chem.* **1992**, *96*, 10898–10905; *Chem. Sci.*, **2016**, *7*, 5758–5764). Compared with the CoPc reference, Co-N-C is obviously lower in pre-edge peak intensity (feature b), indicating partial disruption of the planar central symmetry upon heat treatment. The observation is consistent with the XRD results. The XRD diffraction peaks over Co-N-C precursor before and after calcination at 400 °C are similar to those of the CoPc reference, which corresponds to the characteristic peaks of CoPc standard (PDF:00-044-1995). No obvious peaks assignable to CoPc could be observed after the Co-N-C precursor was calcined at 500 °C or 600 °C, suggesting destruction of the CoPc structure, and Co is highly dispersed on the N-C support in sub-nanoscale (*J. Am. Chem. Soc.* **2018**, *140*, 4218–4221). Nonetheless, there are slight variations in terms of peak shape and strength in transition c and d between the two samples. On the basis of the above analysis, we believe that the structure of Co-N-C is similar to that of CoPc, but there are also some differences.

Figure S3 XRD patterns of various CoPc reference, Co-N-C precursor before and after calcinated at 400, 500 and 600 °C under Ar flow for 2 h.

If XPS finds a valence state of 2+ and 3+ for cobalt, that is not saying much, because XPS is a surface technique, and at the surface you may have oxidized species.

Response: It is possible that the co-existence of 2+ and 3+ valence states for cobalt species on the surface of the Co-N-C catalyst could be originated from surface oxidation. To answer this query, we performed XPS Ar⁺ etching experiments over the Co-N-C and CoPc samples and recorded the spectra before and after Ar⁺ etching (ion energy = 1000 eV). The results are provided in Figure S6.

Before Ar⁺ etching, the binding energies (BEs) of Co2p_{3/2} peak of fresh CoPc and Co-N-C are 780.9 eV and 780.8 eV, respectively, which are higher than those of Co⁰ (778.1 eV) and Co²⁺ (779.2 eV), and only slightly lower than the BE of Co³⁺ (781 eV), suggesting that the dominant states of Co in both fresh samples are +3 (Figure S6a).

After Ar⁺ etching for 60 s under vacuum (Figure S6b), the BE of the Co2p_{3/2} peak of CoPc is 779.05 eV, which is close to that of Co²⁺ species (779.20 eV). As for Co-N-C, the binding energy of the Co2p_{3/2} peak is 780.7 eV, suggesting that the dominant state of Co is still +3. The corresponding discussions have been provided in the main text, see page 6. The related reference has been added as Ref. 10. Meanwhile, we have provided all the XPS results in Figure S6.

Figure S6 XPS Co 2p spectra of (a) fresh Co-N-C and CoPc reference, and (b) Co-N-C and CoPc reference after Ar⁺ etching.